# Extent of N-terminus exposure of monomeric alpha-synuclein determines its aggregation propensity

Amberley D. Stephens [1,8], Maria Zacharopoulou[1,8], Rani Moons[2], Giuliana Fusco[3], Neeleema Seetaloo[4], Anass Chiki[5], Philippa J. Woodhams[1], Ioanna Mela[1], Hilal A. Lashuel[5], Jonathan J. Phillips[4], Alfonso De Simone[6], Frank Sobott[2,7] & Gabriele S. Kaminski Schierle [1✉]

As an intrinsically disordered protein, monomeric alpha-synuclein (aSyn) occupies a large conformational space. Certain conformations lead to aggregation prone and non-aggregation prone intermediates, but identifying these within the dynamic ensemble of monomeric conformations is difficult. Herein, we used the biologically relevant calcium ion to investigate the conformation of monomeric aSyn in relation to its aggregation propensity. We observe that the more exposed the N-terminus and the beginning of the NAC region of aSyn are, the more aggregation prone monomeric aSyn conformations become. Solvent exposure of the N-terminus of aSyn occurs upon release of C-terminus interactions when calcium binds, but the level of exposure and aSyn's aggregation propensity is sequence and post translational modification dependent. Identifying aggregation prone conformations of monomeric aSyn and the environmental conditions they form under will allow us to design new therapeutics targeted to the monomeric protein.

[1] Department of Chemical Engineering and Biotechnology, University of Cambridge, Philippa Fawcett Drive, Cambridge, UK. [2] Department of Chemistry, University of Antwerp, Antwerp, Belgium. [3] Department of Chemistry, University of Cambridge, Lensfield Road, Cambridge, UK. [4] Living Systems Institute, University of Exeter, Stocker Road, Exeter, UK. [5] Laboratory of Molecular and Chemical Biology of Neurodegeneration, Brain Mind Institute, School of Life Sciences, Ecole Polytechnique Fédérale de Lausanne, CH-1015 Lausanne, Switzerland. [6] Department of Life Sciences, Imperial College London, London, UK. [7] School of Molecular and Cellular Biology and The Astbury Centre for Structural Molecular Biology, University of Leeds, Woodhouse Lane, Leeds, UK. [8] These authors contributed equally: Amberley D. Stephens, Maria Zacharopoulou. ✉email: gsk20@cam.ac.uk

In Parkinson's disease (PD) and other synucleinopathies, the monomeric protein alpha-synuclein (aSyn) becomes destabilised, misfolds and aggregates into insoluble, highly structured and β-sheet containing fibrils which form part of Lewy bodies (LB) and Lewy neurites (LN)[1,2]. In its monomeric form, the 14.46 kDa aSyn is a soluble, intrinsically disordered protein (IDP) that is highly flexible and thereby enables plasticity in its function. In particular, it has been proposed that aSyn plays a role in synaptic vesicle recycling and homeostasis in neurons[3,4]. Transient and dynamic electrostatic and hydrophobic intramolecular interactions maintain aSyn in its soluble monomeric form. These intramolecular interactions are responsible for aSyn's smaller radius of gyration ($R_g$) than expected for a 140 residue fully unfolded protein and suggest that some residual structure remains[5]. Therefore, the word 'monomer' actually describes a plethora of conformational states which are constantly reconfiguring. These dynamic interactions are heavily influenced by the surrounding environment of aSyn and their disruption can lead to skewing of the ensemble of monomeric conformations[6]. This, in turn, may influence which aggregation competent/incompetent pathways are taken and whether these are subsequently toxic or not. Identifying conformations of aggregation prone monomeric aSyn or the environment that can destabilise monomeric conformations to favour aggregation will aid in the design of anti-aggregation therapeutics to stabilise the native aSyn conformation.

aSyn is a characteristic IDP with high opposing charge at its termini and low overall hydrophobicity[5]. Monomeric aSyn has three characteristic main regions; the N-terminus, residues 1–60, which is overall positively charged, the non-amyloid-β component (NAC) region 61–95, which is hydrophobic and forms the core of fibrils during aggregation[7], and the C-terminus, residues 96–140, is a highly negatively charged region which binds metal ions[8] (Fig. 1a). To date, six disease-related mutations have been identified in the *SNCA* gene, encoding the aSyn protein, A30P, E46K, H50Q, G51D, A53T, A53E, which are a hallmark for hereditary autosomal dominant PD and are primarily linked to early age, but also late age of onset (H50Q)[9–15]. However, genetic mutations and multiplications of the *SNCA* gene and other PD-associated genes only account for 5–10% of PD cases and the remaining cases are sporadic (idiopathic) and age-related[16]. Yet, we still have not identified mechanistically why these mutations lead to early-onset PD, or what triggers misfolding of wild-type (WT) aSyn.

Intramolecular long-range interactions of aSyn have been detected between many different regions of aSyn. Electrostatic interactions, mediated by the positively charged N-terminus and negatively charged C-terminus, as well as hydrophobic interactions between some residues of the C-terminus and NAC region of aSyn, have been identified using a range of techniques including different nuclear magnetic resonance (NMR) techniques, mass spectrometry (MS) and hydrogen-deuterium exchange MS (HDX-MS)[6,17–25] (Fig. 1b). The importance of these long-range interactions was demonstrated in studies in which charge and hydrophobicity of the protein were altered by mutations, particularly at the C-terminus, leading to differences in the aggregation propensity of aSyn[26–29]. Reduction of charge also occurs during the binding of metal ions, salt ions or polyamines which leads to shielding of the charged N- and C-termini and which permits more energetically favourable packing into fibrils[8,30].

Furthermore, post-translational modifications (PTM), such as nitration and phosphorylation, also alter aggregation rates of aSyn. In particular, phosphorylation of S129 which increases the negative charge of the C-terminus by the addition of a $PO_4^{2-}$ group seems to be pertinent in disease as only 4% of monomeric aSyn is phosphorylated, yet 96% of aSyn in LB and LN are phosphorylated[31]. However, it is not clear whether phosphorylation of S129 is involved in the physiological or pathological function of aSyn, whether it enhances aggregation[32,33] or retards aggregation[34]. In terms of disease association, the presence of aSyn familial mutations leads to different aggregation rates dependent on the mutation. NMR experiments have shown that C-terminus residues are transiently in contact with all six mutation sites at the N-terminus via long-range interactions[23], yet the different mutations lead to differences in levels of solvent exposure, destabilisation, perturbation of the ensemble of conformers and alterations in long-range interactions[35–38]. Identifying conformations or significant long-range interactions that maintain soluble aSyn is critical in understanding what triggers aSyn misfolding, but the difficulty in identifying these long-range interactions and determining the influence of mutations and aggregation prone conformations of aSyn lies in the complexity of sampling an ensemble of dynamic conformations of the monomeric protein.

In the current study, we apply a plethora of techniques, including NMR, HDX-MS, and native nano electrospray ionisation (ESI) MS to study the differences in the conformation of aggregation prone and non-aggregation prone monomeric aSyn including phosphorylated aSyn, pS129, and familial aSyn mutants, A30P, E46K, H50Q, G51D, A53T and A53E. To investigate potential differences in residual structure and long-range interactions, we also performed experiments in the presence of calcium to purposefully skew the dynamic ensemble of conformations as calcium binds at the C-terminus and leads to charge neutralisation[39]. Calcium has been shown to play a role in

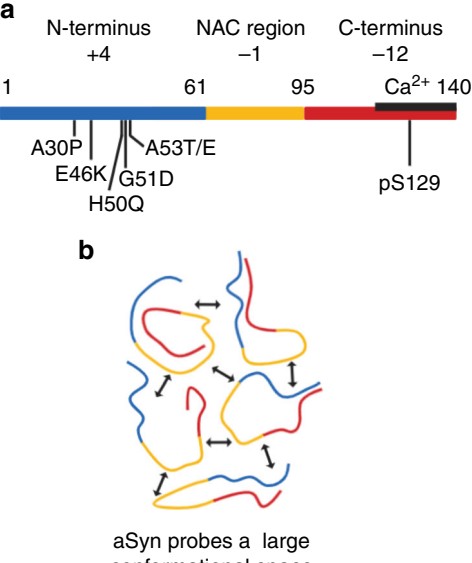

**Fig. 1 Representation of the regions of monomeric aSyn. a** Monomeric aSyn is defined by three regions, the N-terminus, residues 1–60 (blue) with an overall charge of +4, contains the familial mutations A30P, E46K, H50Q, G51D, A53E and A53T. The non-Amyloid-β component (NAC) region, residues 61–95 (yellow), has an overall charge of −1, is highly hydrophobic and forms the core of fibrils. The C-terminus, residues 96–140 (red), is highly negatively charged with an overall charge of −12. Residue S129 is commonly phosphorylated (pS129) in Lewy bodies, but rarely in its soluble state. The calcium binding region (black line) is also found at the C-terminus and spans residues 115–140. **b** Monomeric aSyn is highly dynamic and visits a large conformational space. Transient intramolecular interactions between the N-terminus (blue) and C-terminus (red) and NAC region (yellow) maintain it in a soluble form. Created with BioRender.com.

the physiological and pathological function of aSyn, as calcium binding at the C-terminus of aSyn facilitates interaction with synaptic vesicles and enhances aSyn aggregation rates[8,40]. Furthermore, calcium-aSyn interaction is physiologically relevant as calcium buffering becomes dysregulated in PD and an increase in cytosolic calcium occurs[41]. We therefore also investigated a panel of C-terminus mutants D115A, D119A and D121A which are within the calcium binding region[40].

We conclude that the perturbation of long-range interactions upon calcium binding to monomeric aSyn leads to an increase in N-terminus solvent exposure for some aSyn variants which correlates with their increased aggregation propensity. The extent of N-terminus exposure is influenced by the presence of different PTMs and familial mutations, where the distribution of monomeric conformations of the aSyn ensemble is different between the aSyn variants. The finding that different structural conformations can be identified as early as at the monomer level will be crucial in aiding the development of aSyn aggregation inhibitors stabilising native non-aggregation prone structures.

## Results

**pS129 and D121A have altered conformations compared to WT aSyn**. We first investigated whether interactions mediated by the C-terminus of aSyn were important in modulating monomer conformation and aggregation propensity. We compared WT aSyn to post-translationally modified aSyn, phosphorylated at residue serine 129 (pS129), which increases the C-terminus negative charge, and to a mutant with reduced charge by mutating aspartate (D) to alanine (A) at residue 121 (D121A). The D121A aSyn mutant was chosen from a panel of C-terminus D to A mutants, 115, 119 and 121, which reside in the region of divalent and trivalent cation binding sites[8,42]. D121A aSyn was chosen for further investigation as it displayed a decreased aggregation rate using a thioflavin-T (ThT) based kinetic assay compared to D115A, D119A and WT aSyn (Supplementary Figs. 1–2, Supplementary Note 1).

We first compared the conformation of monomeric D121A and pS129 aSyn to WT aSyn using $^1$H-$^{15}$N heteronuclear single quantum correlation (HSQC) spectra in solution. In comparison to WT aSyn there are chemical shift perturbations (CSPs) around the location of the D121A mutation and around the phosphorylated S129 residue, but otherwise no clear CSPs in the N-terminus and NAC regions (Fig. 2a, Supplementary Fig. 3).

As CSPs of D121A and pS129 aSyn compared to WT aSyn were relatively small aside from the mutation/phosphorylation sites, we next investigated whether addition of calcium could skew the heterogeneous conformational ensemble of monomeric aSyn leading to more homogeneous calcium bound structures which may indicate differences in the structure of the three monomeric aSyn more clearly. Binding of calcium alters electrostatic interactions as it neutralises the negative charge at the C-terminus of aSyn[8]. We performed a calcium titration from 0.2–4.2 mM and observed significant CSPs at the C-terminus for all three aSyn samples (using a fixed protein concentration of 200 μM), as shown previously for WT aSyn (Fig. 2b, Supplementary Fig. 4)[40]. To show that C-terminus CSPs observed in the above experiment are due to a direct interaction of calcium and aSyn and not simply due to global electrostatic effects, we also performed NMR experiments in the presence of 4 mM NaCl. The results in Supplementary Fig. 5 show that we observed no specific C-terminus CSPs. For D121A aSyn, in addition to the main CSPs at the calcium binding region, we observe higher CSPs across the region of residues 1–100 when compared to WT and pS129 aSyn, which is significant as a point mutation usually alters primarily a very localised region of the sequence in a disordered

protein[43] (Fig. 2b, green). Furthermore, we previously observed broadening of the NAC region for WT aSyn when calcium was bound[40], yet we observe no broadening in D121A aSyn (Fig. 2c, Supplementary Fig. 6), suggesting that interactions with the NAC region were altered in D121A aSyn. Our data further indicate that pS129 aSyn also has altered long range interactions in comparison to WT aSyn. Firstly, there appear to be more localised C-terminus CSPs upon calcium binding compared to D121A and WT aSyn (Fig. 2b, yellow). Residues involved in metal binding have previously been shown to be altered upon phosphorylation[42]. Secondly, as observed for D121A aSyn, there was no broadening of the NAC region when calcium was bound (Fig. 2c, Supplementary Fig. 7). These alterations in CSPs upon calcium binding suggest that long range interactions of pS129 and D121A aSyn are already altered compared to WT aSyn which may have implications in terms of the aggregation propensity of these aSyn variants both in the absence and presence of calcium.

In order to determine whether the above structural changes we had observed in the presence of calcium were simply due to changes in the affinity for calcium of the different aSyn variants, we determined the dissociation constant using two different fitting algorithms from the titration experiments. Using a previously applied model[44], but this time using a fixed ligand (calcium) number of 3, we obtained a $K_D$ of 95 (±16) μM, 91 (±16) μM and 69 (±8) μM for WT, pS129 and D121A aSyn, respectively. Using the Hill equation, which also takes into account the level of cooperativity, we obtain a $K_D$ of 670 (±50) μM, 670 (±30) μM and 460 (±30) μM for WT, pS129 and D121A aSyn, respectively. For all fittings, we get an $n > 1$, indicating that calcium binds cooperatively to aSyn (Supplementary Fig. 8).

We next performed ThT-based kinetic assays to investigate whether the above observed conformational differences or calcium binding capacities of aSyn influenced the aggregation propensity of the three aSyn variants. Upon binding to β-sheet rich fibrillary structures the molecule ThT emits fluorescence which provides a read out for the kinetics of aSyn aggregation. The results of the assay showed that although D121A and pS129 aSyn had different charges at the C-terminus, both had a lower aggregation propensity than WT aSyn, particularly in the absence of calcium (Supplementary Fig. 9). The presence of calcium increased the aggregation rate of D121A and pS129 aSyn in comparison to aggregation rates without calcium, however not to the same extent as the rate of WT aSyn. This was also reflected in the concentration of remaining monomer determined by size-exclusion chromatography using high-pressure liquid chromatography (SEC-HPLC) (Supplementary Figs. 9 and 10). It appears that the affinity of aSyn to calcium does not influence aggregation rate per se as the affinity of D121A aSyn for calcium is higher than of WT and pS129 aSyn. We hypothesised that changes in the monomeric long-range interactions may influence aggregation propensity and explain the differences we had observed.

**D121A and pS129 aSyn are less exposed at the N-terminus**. We further investigated whether the perturbations observed by NMR led to region specific shielding or exposure of D121A and pS129 aSyn monomers using HDX-MS. This technique probes the submolecular structure and dynamics of proteins by employing hydrogen-deuterium exchange, and thus permits the identification of protein sequences that are more exposed to the solvent and/or less strongly hydrogen-bonded (deprotected) at physiological pH. Binary comparison of the deuterium uptake profile between WT and D121A aSyn and WT and pS129 aSyn showed that both variants are not significantly different to WT aSyn in terms of exposure (Fig. 3a, b, Supplementary Tables 1 and 2), but this technique may not yet be sensitive enough to determine

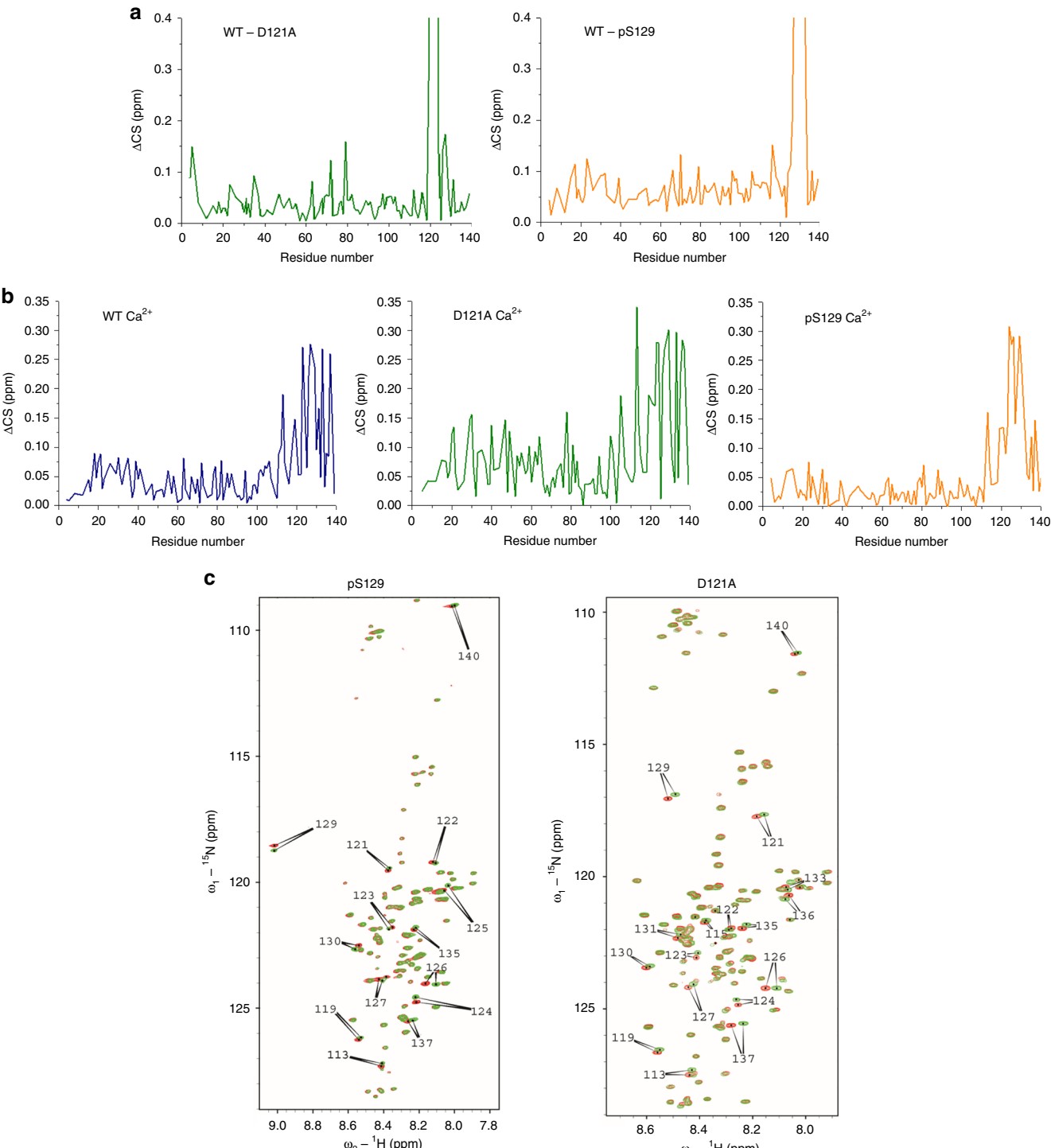

**Fig. 2 ¹H-¹⁵N spectra of WT, D121A and pS129 aSyn indicate conformational differences. a** We compared chemical shift perturbations in the amide backbone of ¹H-¹⁵N D121A aSyn to WT aSyn (green) and pS129 aSyn to WT aSyn (yellow) in 20 mM Tris pH 7.2 (using a fixed protein concentration of 200 μM) and observed CSPs around the location of the D to A mutation and the $PO_4^{2-}$ of S129 at the C-terminus (for the full zoomed out spectrum see Supplementary Fig. 3.). (**b**) Significant CSPs were observed at the C-terminus upon addition of 4.2 mM calcium for all, WT (blue), D121A (green) and pS129 (yellow) aSyn. (Titration data for all concentrations of calcium are available in Supplementary Fig. 4). Higher CSPs are observed for D121A compared to WT and pS129 aSyn. **c** ¹H-¹⁵N HSQC NMR spectra of pS129 and D121A aSyn in the absence (red) and in the presence of calcium (green) (Supplementary Figs. 6 and 7 show spectra with more labels). Major CSPs in the presence of calcium are located at the C-terminus (arrows with assigned amino acid residues) in both pS129 and D121A aSyn.

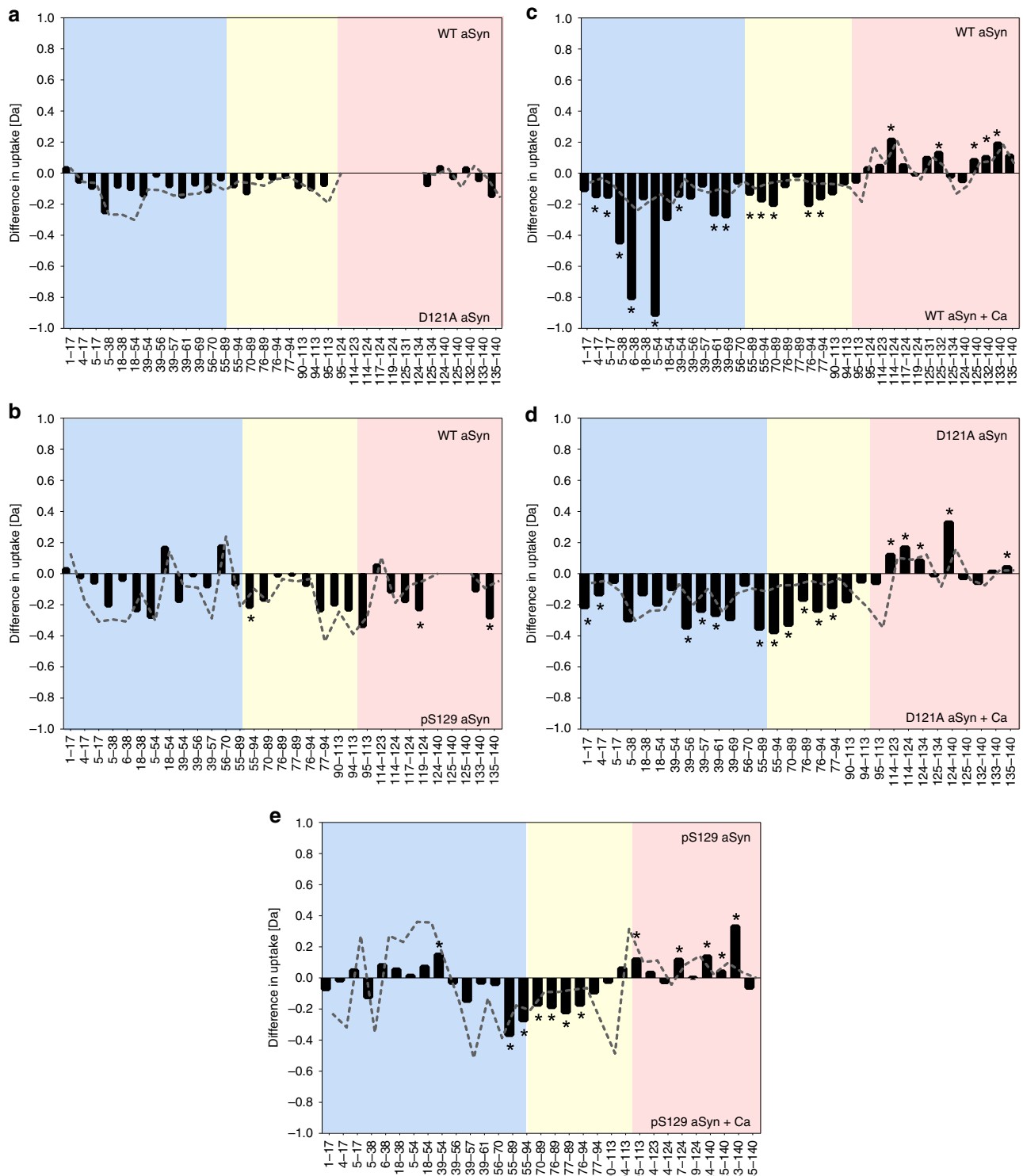

changes in the ensemble of conformations as it is an averaging measurement technique.

We again used calcium to perturb the ensemble of conformations in order to compare alterations of long-range interactions between the three aSyn variants. Binary comparison of the deuterium uptake profile of monomeric WT aSyn revealed solvent protection at the C-terminus and significant deprotection at the NAC and the N-terminus region of aSyn in the presence of calcium compared to the absence of calcium (Fig. 3c,

Supplementary Table 3). This indicates that, when calcium is bound to aSyn, there is reduced exposure to the solvent or increased hydrogen bonding at the C-terminus of aSyn, where calcium binds, as observed by CSPs using NMR, and deprotection of the N-terminus and NAC region. A similar behaviour was observed for D121A and pS129 aSyn as, upon calcium binding, solvent protection at the C-terminus and deprotection at the NAC region of aSyn (Fig. 3d, e, Supplementary Tables 4 and 5). However, while D121A aSyn has a solvent exposed N-terminus,

**Fig. 3 HDX-MS reveals different conformations of D121A and pS129 compared to WT aSyn.** Bars represent differences in deuterium uptake of peptides along the sequence of differently compared aSyn variants (e.g. WT vs D121A aSyn) with the N-terminus region in blue, the NAC region in yellow, and the C-terminus of aSyn in red. Negative values represent increased deuterium uptake in the variant (**a**, **b**) or in the calcium bound state (**c–e**), correlating to more solvent exposure, and less hydrogen bonding. The start and end of each peptide is marked on the x-axis (see aSyn peptide map in Supplementary Fig. 11). Peptides containing the mutation were not comparable to WT aSyn and were removed from the data set, indicated by blank regions. Comparison of the deuterium uptake (in Dalton -Da) between **a** WT and D121A aSyn and **b** WT and pS129 aSyn showed no significant differences to WT aSyn. **c** In the presence of calcium, WT aSyn becomes significantly more deprotected (more solvent exposed/less hydrogen bonded) at the N-terminus and the NAC region, and at the same time becomes solvent protected at the C-terminus. **d** D121A aSyn is significantly more deprotected at the N-terminus and the NAC region upon calcium addition and solvent protected at the C-terminus. **e** pS129 aSyn is deprotected at the NAC region upon calcium addition and solvent protected at the C-terminus, while no significant differences were observed at the N-terminus. The grey dashed line signifies the error (1 s.d.) of six replicates collected per condition. Data acquired at each peptide were subjected to an unpaired Student's $t$-test with alpha set to 1%. Each row was analysed individually, without assuming a consistent SD, individual two-tailed p values are presented in Supplementary Tables 1–5 and significant differences where $p$-values are ≤0.01 are presented by a *. Individual replicate values for deuterium uptake are presented in Supplementary Figs. 22–24.

pS129 aSyn displays little difference in protection levels at the N-terminus. Yet. both D121A and pS129 aSyn, have a more protected/less exposed N-terminus compared to WT aSyn in the calcium-bound state in these averaging measurement data (Fig. 3c–e). Both NMR and HDX-MS indicate that D121A and pS129 aSyn have a different ensemble of conformations compared to WT aSyn. We observe a correlation between the exposure of the N-terminus, which is much less pronounced in D121A and pS129 aSyn than in WT aSyn, with a reduced aggregation propensity in the calcium-bound state, as determined by the ThT fluorescence kinetic assay. This highlights the strength of our approach to determine conformational changes already in monomeric aSyn.

**Aggregation propensity correlates with N-terminus exposure.** To further investigate whether differences in the sub-molecular structure are apparent in the familial aSyn mutants A30P, E46K, H50Q, G51D, A53T and A53E and whether this can influence aggregation rates, we first studied their aggregation kinetics in the presence and absence of calcium using ThT-based kinetic assays. Comparison of the fibrillisation rates of the familial mutants with WT aSyn in the absence of calcium shows that the aSyn mutants A53T, E46K, and H50Q aggregate faster than WT aSyn while the familial aSyn mutants A30P, A53E, and G51D aggregate more slowly than WT aSyn (Fig. 4a, c–e, plate repeats are shown in Supplementary Fig. 12a). Upon the addition of calcium, WT aSyn nucleation and elongation is enhanced, as previously shown[40], and for the fast aggregating aSyn mutants, A53T, E46K, and H50Q the aggregation rate is also enhanced upon the addition of calcium, similarly to WT aSyn. However, the slow aggregating aSyn mutants A30P, A53E, and G51D are either insensitive to calcium addition or aggregate more slowly (Fig. 4b–e, plate repeats are shown in Supplementary Fig. 12b).

We next performed native nano ESI-MS to determine whether differences in the number of calcium ions bound to each aSyn mutant could result in differences in the aSyn aggregation rate. There were no significant differences in the number of calcium ions bound to any of the aSyn mutants, with on average two to three $Ca^{2+}$ ions bound at a 1:10 protein to calcium ratio, and up to 10 ions at high (1:250) calcium concentrations (Fig. 5, Supplementary Figs. 13–15). We also performed native nano ESI-IM-MS in an attempt to resolve heterogeneous structures in the ensemble of aSyn conformations which was not possible with either NMR or HDX-MS. The ion mobility of a protein is determined by the number of collisions the protein ions have with gas molecules, influencing the drift (or arrival) time which is a direct correlation to the rotationally averaged size and shape of the particle. This technique may permit us to determine whether there is a change in the distribution of conformations between different aSyn mutants. In the gas-phase, WT aSyn was found to

have four main conformational distributions at the 8+ charge state which have been observed previously[45–47], however, there were no clear alterations in the distribution of conformations between the mutants and WT aSyn in the presence or absence of calcium or at other charge states. This is discussed further in Supplementary Note 2 (Supplementary Figs. 16–18, Supplementary Tables 6 and 7).

To gain more detailed and localised structural information of the monomeric aSyn mutants we again employed HDX-MS. The familial aSyn mutants A53T and A53E were selected from the familial mutants' panel for this analysis, as they displayed different aggregation behaviour, with faster aggregation rates for A53T and slower aggregation rates for A53E, even though the point mutations are at the same residue. Binary comparison of WT and A53T aSyn (Fig. 6a, Supplementary Table 8), WT and A53E aSyn (Fig. 6b, Supplementary Table 9) and A53T and A53E aSyn (Fig. 6c, Supplementary Table 10) in the absence of calcium showed that there was no significant difference in deuterium uptake between the two compared protein states. Upon the addition of calcium, solvent protection is observed at the C-terminus of A53T and A53E aSyn, similar to WT and the other aSyn variants (see D121A and pS129 aSyn), indicating that the protection at this region is primarily due to calcium binding. At the same time, A53T aSyn is significantly deprotected at the N-terminus, at a similar level to WT aSyn, indicating a breaking of hydrogen bonding or a local unfolding event (Fig. 6d, e, Supplementary Table 11). In contrast, for A53E aSyn (Fig. 6f, Supplementary Table 12) there are mostly no significant differences at the NAC region or at the N-terminus (with the exception of peptides 55–89 and 55–94), upon calcium binding. Thus, the extent to which structural dynamics are perturbed in the N-terminal region in response to binding of calcium at the C-terminus correlates with an increase in the aggregation propensity of aSyn and its variants.

## Discussion
As an intrinsically disordered protein, aSyn is constantly sampling a large conformational space and exists as an ensemble of conformations. The distribution of these conformations is significantly influenced by both changes in the sequence of the protein (e.g. mutations or PTMs) and the surrounding environment. Some of these structural conformers are expected to follow different aggregation pathways, possibly resulting in different fibril polymorphs and even leading to different disease outcomes. By understanding how genetic and environmental factors, such as mutations, PTMs and calcium, influence the dynamics of conformation and favour monomeric aggregation-prone structures, we may begin to understand the initiation of misfolding pathways leading to different synucleinopathies, and subsequently how to disrupt them. However, here within lies the difficulty as the

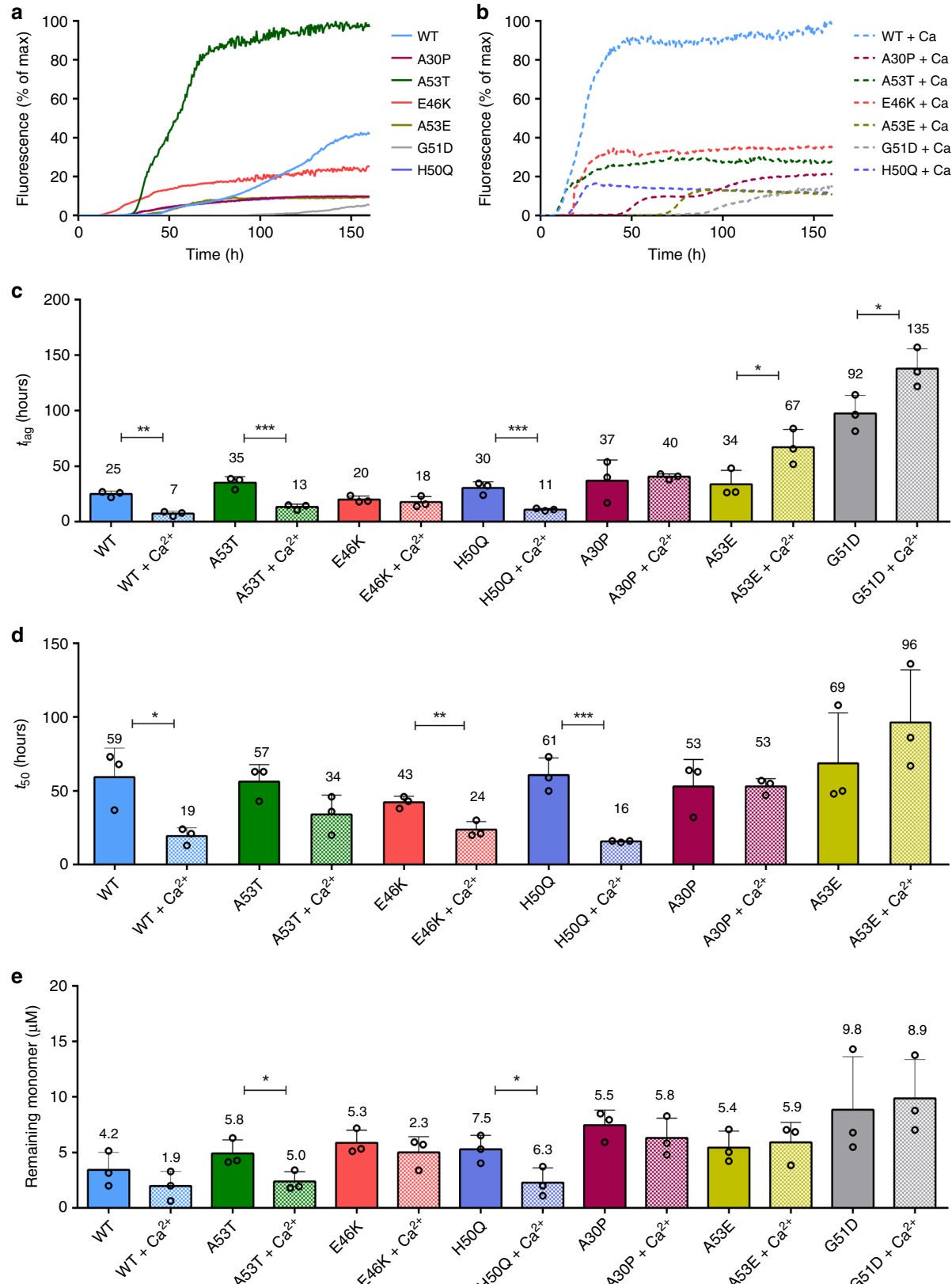

conformations that monomeric aSyn samples are similar in size, charge and structure and are thus hard to detect using ensemble measurement techniques. Furthermore, the presence of a familial mutation does not appear to significantly alter the conformation compared to WT aSyn as observed by these techniques. In this study, we used the biologically relevant ion, calcium, to perturb the conformational ensemble of aSyn structures and compared the differences in CSPs and hydrogen bonding/solvent exposure between aSyn and its mutants. Calcium has been shown to enhance the aggregation rate of WT aSyn[40,48], likely by a similar mechanism as low pH, whereby the reduction of the negative charge at the C-terminus leads to C-terminal collapse when

**Fig. 4 ThT-based aggregation assays reveal different aggregation behaviour for aSyn familial mutants.** Aggregation kinetics of aSyn WT and familial mutants A30P, A53T, E46K, A53E, H50Q, G51D were measured using ThT fluorescence intensity and plotted as % of maximum fluorescence **a** in the absence of calcium and **b** in the presence of 2.5 mM CaCl$_2$ (plate repeat data are available in Supplementary Fig. 12). 20 μM aSyn was incubated with 20 μM ThT in a half area 96 well plate with orbital agitation at 300 rpm for 5 min before each read every hour for 150 h. **c** Lag time ($t_{lag}$) and **d** time to reach 50% of maximum aggregation ($t_{50}$) were calculated and the mean is numerically shown ($t_{lag}$: *$p = 0.0145$, **$p = 0.0034$, ***$p = 0.001$, $t_{50}$: **$p = 0.001$, ***$p = 0.0003$, ***$p = 0.0004$, *$p = 0.043$, *$p = 0.002$)). Mutant G51D did not reach the elongation phase in the studied timeframe, so the $t_{50}$ value was not calculated in (**c**). **e** The remaining monomer concentration was determined using SEC-HPLC, 35 μL of monomer from each well in the ThT assays was analysed on an AdvanceBio SEC 130 Å column in 20 mM Tris pH 7.2 at 1 mL min$^{-1}$. Remaining monomer concentrations were measured from the area under the peak and calculated using a standard curve of known concentrations. The mean remaining monomer concentration is numerically shown (*$p = 0.0438$, *$p = 0.0466$). Measurements were repeated using at least four sample replicates in three experiments and an unpaired two-tailed $t$-test with a 95% confidence interval was used to calculate statistical significance between samples before and after the addition of calcium. Error bars represent s.d.

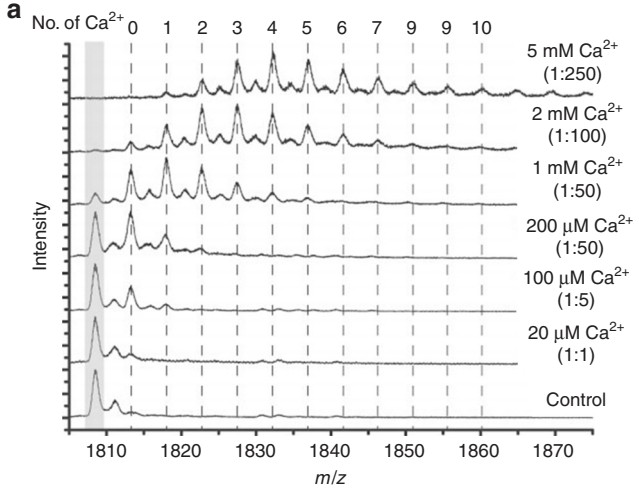

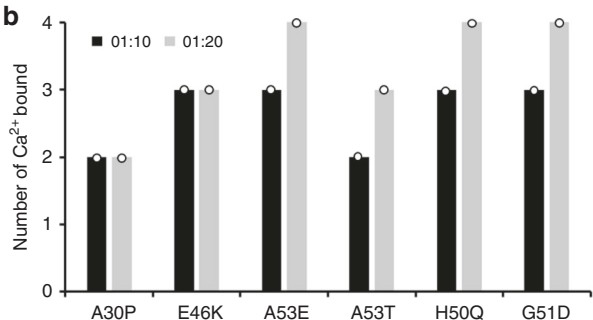

**Fig. 5 Native nano ESI-MS shows no significant difference in number of calcium ions bound. a** A calcium titration from 20 μM to 5 mM shows increasing numbers of calcium ions bound to WT aSyn in the +8 charge state (20 μM in 20 mM ammonium acetate) with a maximum of 10 calcium ions bound at a 1:250 ratio (the full spectrum is displayed in Supplementary Fig. 13). **b** At ratios of 1:10 (black) and 1:20 (grey) aSyn to calcium a maximum of two to four calcium ions are bound to aSyn mutants (mass spectra are displayed in Supplementary Fig. 14). The number of calcium ions bound in three replicates was the same for all aSyn mutants, as displayed by the circle, therefore, no error is displayed.

electrostatic repulsion is reduced, leading to altered long-range electrostatic interactions and enhanced hydrophobic interaction between the C-terminus and the NAC region which drives aggregation[22,49–52]. We observed mutant specific differences in long-range interactions, compaction and solvent exposure compared to WT aSyn which correlate to aggregation propensity and will be discussed individually.

We first investigated the role of charge and long-range interactions at the C-terminus by comparing two aSyn variants with reduced (D121A) and added (pS129) negative charge. We observed that both D121A and pS129 aSyn displayed reduced aggregation rates in comparison to WT aSyn, indicating that increasing or decreasing charge by one residue at the C-terminus does not decrease or increase aggregation rates, respectively, as it may have been expected. We observe that calcium binding affinity does not correlate with aggregation rates, suggesting that aggregation rates are not altered by a single residue charge or calcium binding per se. Furthermore, our results show that there is positive cooperativity for calcium binding to aSyn, which indicates that charge and electrostatic interactions are not the driving force for calcium binding, as this would lead to negative cooperativity. We argue that calcium binding leads to a conformational change, which consequently leads to a positive cooperativity. This subsequent conformational change and the extent of the altered conformation governs aggregation propensity. $K_D$ predictions of intrinsically disordered proteins are known to be challenging, in particular as calcium binding to aSyn does not lead to a characteristic conformation-induced protein-ligand binding. The highly dynamic nature of monomeric aSyn may not permit defined calcium binding structures to be formed, making it difficult to extract an exact $K_D$ value. This may explain why the two fitting models vary in their $K_D$ predictions.

We only observed small differences in the conformational properties of monomeric aSyn, as revealed by changes in CSPs and solvent exposure compared to WT aSyn in the absence of calcium for D121A or pS129 aSyn. This is likely due to the vast array of conformations sampled by the protein leading to minimal structural changes using averaging measurement techniques such as NMR and HDX-MS. However, by perturbing the ensemble of conformations, as seen upon the addition of calcium, we could observe differences in CSPs and deuterium uptake upon calcium binding. By NMR, we observe pS129 aSyn has a different calcium-binding region and displays no broadening in the NAC region compared to WT aSyn, while D121A has a higher degree of CSPs across the sequence, but also no NAC region broadening, suggesting long-range interactions with the NAC region in both these variants are altered before calcium binds. Broadening in the NAC region has been observed for WT aSyn when bound to calcium[40] and at low pH, suggesting enhanced interactions between the charge neutralised calcium binding C-terminus and the NAC region[22].

Regarding aSyn's propensity to aggregate, we observe, using HDX-MS, that the N-terminus of pS129 aSyn is not solvent exposed upon calcium binding, whereas for D121A aSyn, although the N-terminus is slightly exposed, is significantly less exposed compared to WT aSyn upon calcium binding, which is correlated with the decreased aggregation propensity of both D121A and pS129 aSyn. Previous HDX-MS studies have

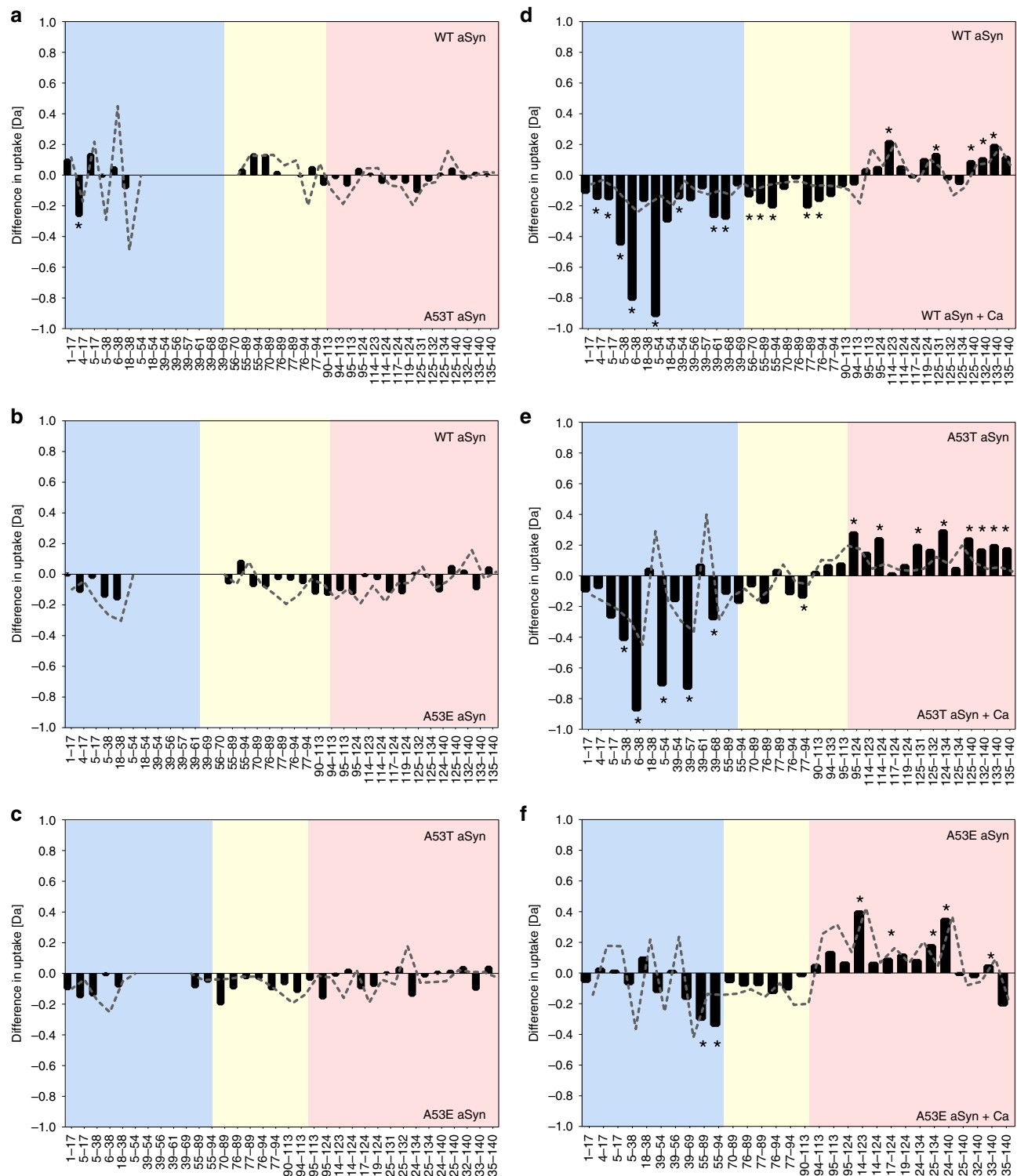

suggested that the N-terminus of aSyn may be involved in aggregation as there is heterogeneity in its solvent exposure during aggregation, while the C-terminus remains completely solvent exposed[53–56]. Yet, we show here that it is the extent of N-terminus exposure that influences aggregation propensity. Importantly, for monomeric aSyn we only observe N-terminus exposure upon disruption of long-range interactions with the C-terminus upon calcium binding. The C-terminus thus greatly influences aggregation propensity, as it is important in maintaining long range interactions and solubility of monomeric

aSyn[57,58]. Many studies investigating the effect of the C-terminus, particularly its charge, on aSyn aggregation rates use C-terminus truncations which heavily disrupt and remove long-range interactions. Instead studying mutations of specific residues gives us further insight into the importance of sequence defined interactions. For instance mutation of proline[59], or glutamate[58] residues to alanine at the C-terminus increases aSyn aggregation propensity, yet mutation of tyrosine[27] residues to alanine decreases aSyn aggregation propensity. Even phosphorylation of S129 compared to Y125 aSyn (only 4 residues apart) leads to

**Fig. 6 HDX-MS reveals different conformations in A53E compared to WT and A53T aSyn.** Bars represent differences in deuterium uptake of peptides along the sequence of differently compared aSyn mutants with the N-terminus region in blue, the NAC region in yellow, and the C-terminus in red. Negative values represent increased deuterium uptake in the mutant (**a–c**) or in the calcium bound state (**d–f**), correlating to more solvent exposure, and less hydrogen bonding. The start and end of each peptide is marked on the x-axis (from aSyn peptide map see Supplementary Fig. 11). Peptides containing the mutation were not comparable to WT aSyn and were removed from the data set, indicated by blank regions. Difference in deuterium uptake (Da) between **a** WT and A53T aSyn, **b** WT and A53E aSyn and **c** A53T and A53E aSyn showed no significant differences throughout the sequence. In the presence of calcium, as previously shown in Fig. 2c, **d** WT aSyn becomes significantly deprotected at the N-terminus and the NAC region, and more solvent protected at the C-terminus. **e** Similarly, A53T aSyn is significantly deprotected at the N-terminus and the NAC region upon calcium addition, and at the same time becomes solvent protected at the C-terminus. **f** A53E aSyn also becomes solvent protected at the C-terminus upon calcium addition but no significant changes are observed at the NAC region and most of the N-terminus. The grey trace signifies the error (1 s.d.) of six replicates collected per condition. Data acquired at each peptide were subjected to an unpaired Student's t-test with alpha set to 1%. Each row was analysed individually, without assuming a consistent SD, individual two-tailed p values are presented in Supplementary Tables 1–5 and significant differences where p-values are ≤0.01 are presented by a *. Individual replicate values for deuterium uptake are presented in Supplementary Figs. 25–27.

differences in aggregation rates, as pY125 does not influence the aggregation rate when compared to WT aSyn[60], while pS129 reduces aggregation rates even in aSyn truncated at Y133 and D135[61]. pY125 aSyn also does not have an altered binding capacity to C-terminus nanobodies nor an altered binding region of metals as pS129 aSyn does, indicating that there are very site-specific functions and interactions present at the C-terminus which warrant further exploration. It further shows that sequence-specific interactions are important in maintaining long-range interactions[42,60]. Although D121A aSyn is not a naturally occurring mutation, by comparing it to pS129, we can explore how altering long-range interactions of monomeric aSyn at the C-terminus by mutation or PTM and variation of the local environment, e.g. addition of calcium, leads to altered interactions with the NAC region and different levels of solvent exposure at the N-terminus. Our results highlight that both sequence and environment are important in determining aSyn's aggregation propensity.

While all familial aSyn point mutations reside at the N-terminus, the putative calcium binding site is at the C-terminus[62]. Certainly, the presence of a familial aSyn mutation has been shown to alter interactions between the mutation site and the C-terminus by NMR compared to WT aSyn, with varying differences apparent for each aSyn mutant in both physiological and mildly acidic conditions[23,36]. The increase in aggregation propensity of WT aSyn upon calcium binding cannot only be explained by charge neutralisation at the C-terminus as all familial aSyn mutants should have responded in the same way to calcium, instead we observe that the familial aSyn mutants have different aggregation kinetics. It is more likely that the difference in aggregation propensity is a result of perturbed long-range interactions with the C-terminus, which is understandable considering the mutation regions interact with the calcium binding regions at the C-terminus. Using HDX-MS, we observe mutant specific differences in the level of solvent protection at the N-terminus of A53T and A53E aSyn upon binding of calcium suggesting that indeed different long-range interactions are present. A local unfolding event (deprotection) at the N-terminus and the NAC region of WT and A53T aSyn correlates with their increased aggregation propensity upon calcium addition. No such deprotection was observed for the aSyn mutant A53E, where no significant differences where observed upon calcium binding, except for two small peptides, and whose aggregation kinetics were, firstly slower than both WT and A53T aSyn, but also had similar kinetics in the presence and absence of calcium. In another study, the 'fast aggregating' mutant, E46K, also displays an increased solvent exposure across its sequence, and in addition, N-terminus residues of E46K aSyn are involved in oligomerisation[36]. We expect that the release of long-range contacts

with the C-terminus leads to exposure of the N-terminus and the NAC region, observed in WT and A53T aSyn, to be a major factor influencing aSyn aggregation kinetics. Previous experiments have shown that the N-terminus of aSyn may modulate aSyn aggregation; altering sequence, cross-linking specific residues and targeting binding proteins to the N-terminus leads to variation in aggregation kinetics[63–68]. Previous studies have shown mutant aSyn have altered long-range interactions compared to WT aSyn[35,69–72]. Our study suggests that the altered stability at the N-terminus combined with mutant aSyn's lower propensity to form an α-helix compared to WT aSyn, may skew the distribution of the conformational ensemble leading to conformations with a high propensity to form oligomers and fibrils. The fact that such differences in aSyn long-range contacts and an increase in N-terminus solvent exposure can be identified as early as at the monomer level, and correlate to the propensity to fibrillise is important, and a step towards understanding early events in the misfolding pathway and how structure and environmental factors may influence aggregation propensity of the monomer. Interestingly the N-terminus and NAC region of aSyn have also been implicated in the formation of phase separated aSyn[73]. It remains to be determined whether the structure of the initial monomer subsequently influences the tertiary structural fate of aSyn or influences fibril structure.

As an IDP, aSyn samples many different conformations making it difficult to identify specific aggregation prone conformations, particularly using ensemble measurement techniques. By comparing the submolecular structure of calcium-bound aSyn variants, we instead sampled a skewed population and inferred differences in structure and aggregation propensity as part of a response to calcium binding at the monomer level. We attribute the increase in aggregation propensity upon calcium binding to structural perturbation, as we observe no correlation in aSyn's aggregation propensity to its affinity to calcium, the number of calcium ions bound or charge neutralisation at the C-terminus. Instead, we observe different responses to calcium based on the presence of different long-range interactions which are likely already altered in non-calcium bound forms of the familial mutants and upon the addition of $PO_4^{2-}$ at S129 (Fig. 7). Calcium binding leads to further disruption of intramolecular interactions with the C-terminus leading to unfolding and solvent exposure of the N-terminus.

The extent of N-terminus solvent exposure upon disruption of contacts with the C-terminus when calcium binds correlates with the aggregation propensity of aSyn; pS129, A53E, and to some extent D121A aSyn, were less solvent exposed at the N-terminus and had a reduced aggregation propensity compared to WT and A53T aSyn which were more aggregation prone and more solvent exposed at the N-terminus and at the beginning of the NAC

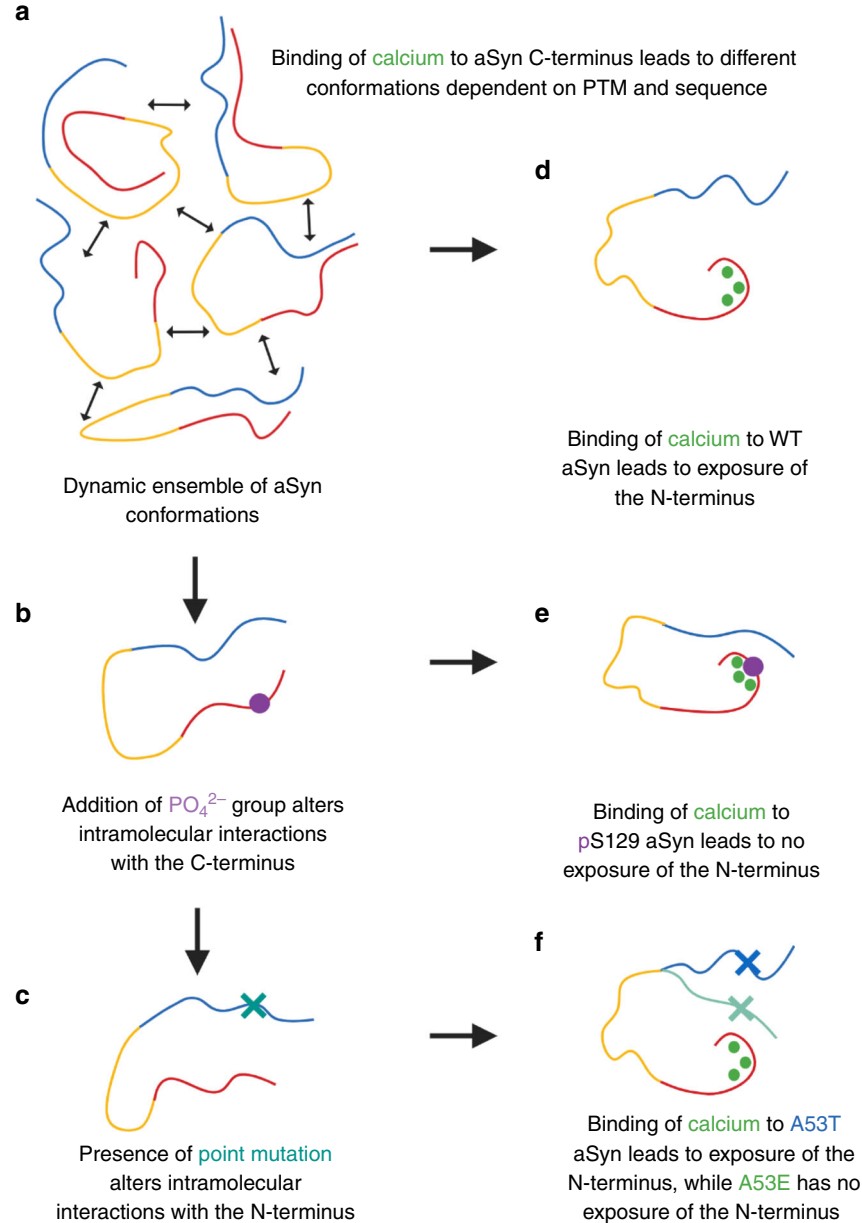

**Fig. 7 Cartoon representation of the effect of phosphorylation, mutation and calcium binding on aSyn. a** aSyn is a dynamic ensemble of conformations in solution. Its solubility is maintained by long-range interactions between the N-terminus (blue), NAC region (yellow) and the C-terminus (blue) of aSyn. **b** Addition of a phosphate group to S129 (purple circle) or **c** a point mutation (teal cross) alters the long-range interactions and skews the dynamic ensemble to favour or disfavour aggregation prone structures. **d** Addition of divalent cations such as calcium (green) leads to charge shielding and exposure of the N-terminus of aSyn. **e** Calcium binding to phosphorylated aSyn leads to different perturbations of long-range interactions. Phosphorylation at S129 leads to a slightly different calcium binding region and to reduced solvent exposure of the N-terminus compared to WT aSyn. **f** The aSyn mutation A53T (blue with blue cross) leads to N-terminus solvent exposure upon calcium binding at the C-terminus, similar to WT aSyn, however, the aSyn mutation A53E (light teal with teal cross) leads to less N-terminus solvent exposure upon calcium binding. Created with BioRender.com.

region (Fig. 7). It remains to be determined whether these different structures of monomeric aSyn can be isolated, potentially by cross-linking to stabilise structures, or whether they have different toxicity in cells. Furthermore, it would be interesting to investigate whether these conformational changes and exposure of the N-terminus lead to a reduction of the energy barrier needed to be overcome for dimer formation, which may be crucial in determining whether aSyn becomes pathological or not. This would be an important step in rationalising the molecular mechanism of PD and other synucleinopathies by identifying which local environmental factors bias the monomeric population towards the most disease-relevant conformers. Finally, targeting the most disease-relevant monomeric structures of aSyn and to convert them into 'normal' functional structures could open up a new chapter in the design of therapeutics against PD and other synucleinopathies.

## Methods

**Purification of aSyn**. Human wild-type (WT) alpha-synuclein was expressed using plasmid pT7-7. aSyn mutations D115A, D119A, D121A, A30P, E46K, A53T, A53E, H50Q and G51D were introduced using the QuikChange Lighting Site-Directed Mutagenesis K (Agilent Technologies LDA UK Limited, UK), primer list is

available on the Cambridge University Repository and Supplementary Table 13, and confirmed by sequencing (Source Bioscience, Nottingham, UK). The plasmids were heat shocked into *Escherichia coli* One Shot® BL21 STAR™ (DE3) (Invitrogen, Thermo Fisher Scientific, Cheshire, UK) and protein expression induced with isopropyl β-d-1-thiogalactopyranoside (IPTG). Recombinant aSyn was purified using ion exchange chromatography (IEX) in buffer A (10 mM Tris, 1 mM EDTA pH 8) against a linear gradient of buffer B (10 mM Tris, 1 mM EDTA, 0.5 M NaCl pH 8) on a HiPrep Q FF 16/10 anion exchange column (GE Healthcare, Uppsala, Sweden). aSyn was then dialysed into buffer C (1 M $(NH_4)_2SO_4$, 50 mM Bis-Tris pH 7) and further purified on a HiPrep Phenyl FF 16/10 (High Sub) hydrophobic interaction chromatography (HIC) column (GE Healthcare) and eluted against buffer D (50 mM Bis-Tris pH 7). aSyn was extensively dialysed against 20 mM Tris pH 7.2 and concentrated using 10 k MWCO amicon centrifugal filtration devices (Merck KGaA, Darmstadt, Germany) and stored at −80 °C until use. Before experiments 1 mL of aSyn was further purified using a Superdex 75 pg 10/300 GL size exclusion chromatography (SEC) column (GE Healthcare) in 20 mM Tris pH 7.2 to obtain monomeric protein. Purification was performed on an ÄKTA Pure (GE Healthcare). Protein concentration was determined by measuring the absorbance at 280 nm on a Nanovue spectrometer using the extinction coefficient 5960 $M^{-1}cm^{-1}$.

Protein purity was analysed using analytical reversed phase chromatography. Each purification batch was analysed using a Discovery BIO Wide Pore C18 column, 15 cm × 4.6 mm, 5 μm, column with a guard cartridge (Supelco by Sigma-Aldrich) with a gradient of 95% to 5% $H_2O$ + 0.1% trifluoroacetic acid (TFA) and acetonitrile + 0.1% TFA at a flow-rate of 1 mL/min. The elution profile was monitored by UV absorption at 220 nm and 280 nm on an Agilent 1260 Infinity HPLC system (Agilent Technologies LDA UK Limited, UK) equipped with an autosampler and a diode-array detector (a representative chromatograph is shown in Supplementary Fig. 20a and a Coomassie blue stained SDS-PAGE gel of monomeric pure aSyn in Fig. 20b). Protein purity fell between 89 and 96%. Figures of the chromatographs were created in and exported from MatLab R2019a (MathWorks, USA).

**Purification of aSyn for nuclear magnetic resonance experiments.** *E. coli* was grown in isotope-enriched M9 minimal medium containing $^{15}N$ ammonium chloride, and $^{13}C$-glucose[74]. To isolate expressed aSyn the cell pellets were lysed by sonication in 10 mM Tris, 200 μM EDTA, protease inhibitor tablets (cOmplete™, EDTA-free, Roche) pH 8. The cell lysate was centrifuged at 20 k × g for 30 min and the supernatant was then heated for 20 min at 90 °C to precipitate the heat-sensitive proteins and subsequently centrifuged at 22 k × g for 30 min to remove precipitated proteins. Streptomycin sulfate (Sigma-Aldrich) 10 mg ml$^{-1}$ was added to the supernatant and incubated at RT for 15 min to precipitate DNA, the mixture was centrifuged at 22 k × g for 30 min to remove DNA and then repeated. Ammonium sulfate 360 mg ml$^{-1}$ was added to the supernatant and stirred for 30 min to precipitate the aSyn protein. The solution centrifuged at 22 k × g for 30 min and the resulting pellet was resuspended in 25 mM Tris-HCl, pH 7.7 and dialysed overnight. The protein was purified by IEX, described above, then further purified by SEC on a HiLoad 16/60 Superdex 75 prep grade column (GE Healthcare) in 20 mM Tris pH 7.2. All the fractions containing the monomeric protein were pooled together and concentrated by using 10 k MWCO centrifugal filtration devices (Merck).

**Purification of phosphorylated serine 129 aSyn.** WT aSyn $^{13}C/^{15}N$-labelled (aSyn $^{13}C/^{15}N$) was expressed and purified as previously described[75,76]. Briefly, *E. coli* BL21(DE3) cells were transfected with a pT7-7 plasmid containing WT aSyn and was cultured in an isotopically supplemented minimal media[75], then aSyn $^{13}C/^{15}N$ was purified using an anion exchange chromatography followed by reversed-phase HPLC (RP-HPLC) purification using a Proto 300 C4 column and a gradient from 30 to 60% B over 35 min at 15 ml/min, where solvent A was 0.1% TFA in water and solvent B was 0.1% TFA in acetonitrile, the fractions containing the protein were pooled and lyophilized and the protein was stored at −20 °C. For the preparation of phosphorylated S129 aSyn $^{13}C/^{15}N$ (aSyn $^{13}C/^{15}N$ pS129), we used a protocol using PLK3 kinase to introduce selectively the phosphorylation at S129[77]. The WT aSyn $^{13}C/^{15}N$ was resuspended in the phosphorylation buffer (50 mM HEPES, 1 mM MgCl$_2$, 1 mM EGTA, 1 mM DTT) at a concentration of ~150 μm and then 2 mM of ATP and 0.42 μg of PLK3 kinase (Invitrogen) per 500 μg of protein were added. The enzymatic reaction was left at 30 °C overnight without shaking. Upon complete phosphorylation, as monitored by mass spectroscopy (LC/MS), aSyn $^{13}C/^{15}N$ pS129 was purified from the reaction mixture by RP-HPLC using an Inertsil WP300–C8 semiprep column. Finally, the fractions containing the protein of interest were pooled and quality control of aSyn $^{13}C/^{15}N$ pS129 was performed using mass spectroscopy, UPLC, and SDS-PAGE, the protein was 99.89% phosphorylated (Supplementary Fig. 20c–e).

**Solution nuclear magnetic resonance (NMR).** In order to probe the structure and thermodynamics of calcium binding with aSyn WT, pS129 and D121A at a residue specific level, we employed a series of $^1H$-$^{15}N$ HSQC experiments using different concentrations of $Ca^{2+}$ (0.0 mM–4.2 mM) and a fixed concentration of aSyn (200 μM). The pH of samples was checked before experiments were

performed and after calcium addition. NMR experiments were carried out at 10 °C on a Bruker spectrometer operating at $^1H$ frequencies of 800 MHz equipped with triple resonance HCN cryo-probe and data collected in the Topspin 3.6.0 software (Bruker, AXS GmBH). The $^1H$-$^{15}N$ HSQC experiments were recorded using a data matrix consisting of 2048 ($t_2$, $^1H$) × 220 ($t_1$, $^{15}N$) complex points. Assignments of the resonances in $^1H$-$^{15}N$-HSQC spectra of aSyn were derived from our previous studies and data analysed using the Sparky 3.1 software.

The perturbation of the $^1H$-$^{15}N$ HSQC resonances was analysed using a weighting function in Eq. 1:

$$\Delta\delta = \sqrt{\frac{1}{2}\left(\delta_H^2 + 0.15\delta_N^2\right)} \qquad (1)$$

The titration enabled calculating the fraction of bound aSyn, $\chi_B$, as a function of $[Ca^{2+}]$.

$$\chi_B = \frac{\Delta\delta_{obs}}{\Delta\delta_{sat}} \qquad (2)$$

Where the $\Delta\delta_{obs}$ is the chemical shift perturbation of the amide group of a residue of aSyn at a given $[Ca^{2+}]$ and $\Delta\delta_{sat}$ is the perturbation obtained with calcium saturation. $\chi_B$ values were obtained as a function of $[Ca^{2+}]$ for every residue of the protein for which resolved peaks in the $^1H$-$^{15}N$ HSQC are available. A global $\chi_B$ was calculated by average the fractions corresponding to residues associated with major resonance perturbations in the presence of calcium.

In order to obtain the apparent dissociation constant, we used two different models.

As first, we employed a model based on previous investigations[40]:

$$\alpha syn^U + (Ca^{2+})_L = \alpha syn^B (Ca^{2+})_L \qquad (3)$$

Where $\alpha syn^U$ and $\alpha syn^B$ indicate free and calcium bound aSyn, $L$ indicates the number of $Ca^{2+}$ interacting with one aSyn molecule, and the overall concentration of aSyn in this equilibrium is given by

$$[\alpha syn] = [\alpha syn^U] + [\alpha syn^B (Ca^{2+})_L] \qquad (4)$$

the apparent dissociation constant in this model corresponds to:

$$K_D = \frac{[\alpha syn^U][Ca_L^{2+}]}{[\alpha syn^B (Ca^{2+})_L]} \qquad (5)$$

Which provides a formula for $\chi_B$:

$$\chi_B = \frac{[\alpha syn] + \left[\frac{Ca^{2+}}{L}\right] + K_D - \sqrt{\left(\left[\alpha syn\right] + \left[\frac{Ca^{2+}}{L}\right] + K_D\right)^2 - \frac{4[\alpha syn][Ca^{2+}]}{L}}}{2[\alpha syn]} \qquad (6)$$

When using this fitting model in the case of WT aSyn we obtained a $K_D$ of 21 (±5) μM and an $L$ of 7.8 (±0.51)[40]. Based on the present MS data, we here fixed the value of $L$ to 3.

We then used a different model that accounts for the cooperativity of the binding. In particular, we used the Hill equation to fit our data in Eq. 7:

$$\chi_B = \frac{[Ca^{2+}]^n}{K_D + [Ca^{2+}]^n} \qquad (7)$$

Where the Hill coefficient, $n$, describes the cooperativity of the binding. A $n$ value higher than 1 indicates a positive cooperativity for the binding.

**Hydrogen-deuterium exchange mass spectrometry (HDX-MS).** Hydrogen exchange was performed using an HDX Manager (Waters, USA) equipped with a CTC PAL sample handling robot (LEAP Technologies, USA). Samples of aSyn (10 μM) in protonated aqueous buffer (20 mM Tris, pH 7.2) were diluted 20-fold into deuterated buffer (20 mM Tris, pD 7.2) at 20 °C, initiating hydrogen exchange. The same was performed for the calcium condition, in protonated and deuterated buffers containing 1 mM CaCl$_2$ (20 mM Tris, pH 7.2, 1 mM CaCl$_2$, and 20 mM Tris, pD 7.2, 1 mM CaCl$_2$). Samples were pre-incubated in calcium containing buffer for 30 min in a aSyn:CaCl$_2$ ratio of 1:100. The protein was then incubated for 30 s in the deuterated buffer and six replicates were collected per condition. Hydrogen exchange was arrested by mixing 1:1 with pre-chilled quench solution (100 mM Tris, 8 M Urea, pH 2.45 at 0 °C). The protein was then digested into peptides on a pepsin column (Enzymate, Waters) and the peptides were separated on a C18 column (1 × 100 mm ACQUITY UPLC BEH 1.7 μm, Waters) with a linear gradient of acetonitrile (3–40%) supplemented with 0.2% formic acid. Peptides were analysed with a Synapt G2-Si mass spectrometer (Waters). The mass spectrometer was calibrated with NaI calibrant in positive ion mode. A clean blank injection was ran between samples to minimise carry-over. Peptide mapping of aSyn, where peptides were identified by MS/MS fragmentation, was performed prior to the hydrogen exchange experiments and analysed using ProteinLynx Global Server- PLGS (Waters). Peptide mapping of aSyn yielded coverage of 100% of aSyn with a high degree of redundancy (5.38) (Supplementary Fig. 11). The following parameters were used to filter the quality of the peptides: minimum and maximum peptide sequence length of 4 and 50, respectively, minimum intensity of 5000, and minimum products per amino acid of 0.2. All spectra generated from the peptides were examined and only peptides with high-quality spectra and a high signal to noise ratio were used for data analysis. The data

pertaining to deuterium uptake (representative data presented in Supplementary Fig. 21 and all replicates presented in Supplementary Figs. 22–27) were analysed and visualised in DynamX 3.0 (Waters) and GraphPad Prism 8 (GraphPad Software, US). No correction was made for back-exchange. To allow access to the HDX data of this study, the HDX data summary table is included in the supporting information as per consensus guidelines[78].

**Thioflavin-T (ThT) based assays**. 10 μM freshly made ThT (Abcam, Cambridge, UK) was added to 50 μL of 10 μM aSyn in 20 mM Tris pH 7.2. All samples were loaded onto nonbinding, clear bottom, 96-well half-area plates (Greiner Bio-One GmbH, Germany). The plates were sealed with a SILVERseal aluminium micro-plate sealer (Grenier Bio-One GmbH). Fluorescence measurements were taken with a FLUOstar Omega plate reader (BMG LABTECH GmbH, Ortenbery, Germany). The plates were incubated at 37 °C with orbital shaking at 300 rpm for five minutes before each read every hour. Excitation was set at 440 nm with 20 flashes and the ThT fluorescence intensity measured at 480 nm emission with a 1300 gain setting. ThT assays were repeated at least 3 times using four or more wells for each condition. Data were normalised to the well with the maximum fluorescence intensity for each plate and the average fluorescence intensity was calculated for all experiments in Microsoft Excel 2016 (Microsoft Office, Washington, USA). The lag time ($t_{lag}$) and half-life of the fluorescence ($t_{50}$) were calculated for each aSyn variant, also in Microsoft Excel. An unpaired two-tailed $t$-test, with a 95% confidence interval, was used to calculate statistical significance between samples before and after the addition of calcium using GraphPad Prism 8.

**Analytical size exclusion chromatography (SEC)**. SEC-HPLC analysis was used to calculate the remaining aSyn monomer concentration in each well at the end of the ThT assays. The contents of each well after the ThT-based assay were centrifuged at $21 k \times g$ for 20 min and the supernatant added to individual aliquots in the autosampler of the Agilent 1260 Infinity HPLC system (Agilent Technologies LDA UK Limited, UK). 35 μL of each sample was injected onto an Advance Bio SEC column, $7.8 \times 300$ mm 300 Å (Agilent, UK) in 20 mM Tris pH 7.2 at 1 mL min$^{-1}$ flow-rate. The elution profile was monitored by UV absorption at 220 and 280 nm. A calibration curve of known concentrations of aSyn was used to calculate the remaining monomer concentration of aSyn in each well. Two or three wells per experiment for three experiments were analysed for the remaining monomer concentration. Figures of the chromatographs were created in and exported from MatLab R2019a (MathWorks, USA).

**Nano electrospray ionisation mass spectrometry (Nano ESI-MS)**. A final concentration of 20 μM aSyn (WT or mutant) was obtained in 20 mM ammonium acetate (Sigma Aldrich, St. Louis, MO, USA) pH 7 and measured as a control. CaCl$_2$ (Merck, Darmstadt, Germany) was dissolved in deionised H$_2$O and added to the sample with a final concentration ranging between 200 μM and 400 μM. The samples were incubated for 10 min at room temperature before measuring. Nano-ESI mass spectrometry measurements were performed on a Synapt G2 HDMS (Waters, Manchester, UK) and analysed using Masslynx version 4.1 (Waters, Manchester, UK). For infusion into the mass spectrometer, home-made gold-coated borosilicate capillaries were used. The main instrumental settings were: capillary voltage 1.4–1.8 kV; sampling cone 25 V; extraction cone 1 V; trap CE 4 V; transfer CE 0 V; trap bias 40 V. Gas pressures used throughout the instrument were: source 1.5–2.7 mbar; trap cell $2.3 \times 10^{-2}$ mbar; IM cell 3.0 mbar; transfer cell $2.5 \times 10^{-2}$ mbar. Data were analysed and exported from Origin version 8 (Origin Lab Corporation, Northampton, USA).

**Reporting summary**. Further information on experimental design is available in the Nature Research Reporting Summary linked to this paper.

## Data availability

Source data underlying Figs. 3, 4, 5, 6, Supplementary Figs. 1, 9, 11–15, 17–19, 21–27 are provided as a Source data file. Other data sets associated with all figures are available from the University of Cambridge Data Repository https://doi.org/10.17863/CAM.50622. Additional figures and data for ThT-based kinetic assays, AFM data, NMR calcium titration data, additional HDX-MS data, additional native MS data, all nano-ESI-IM-MS data, purity of aSyn samples, Supplementary Methods and primer sequences are found in the Supplementary Information.

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

## Acknowledgements

G.S.K.S. acknowledges funding from the Wellcome Trust (065807/Z/01/Z) (203249/Z/16/Z), the UK Medical Research Council (MRC) (MR/K02292X/1), Alzheimer Research UK (ARUK) (ARUK-PG013-14), Michael J Fox Foundation (16238) and Infinitus China Ltd. A. DS. Acknowledges funding from the European Research Council (ERC) Consolidator Grant (CoG) 819644 "BioDisOrder". A.D.S. and M.Z. acknowledge Alzheimer Research UK for travel grants. G.F. acknowledges funding from St John's College (Cambridge). M.Z. acknowledges funding from Newnham College (Cambridge) and the

George and Marie Vergottis Foundation (Cambridge Trust) and the British Biophysical Society (BSS) for travel grants. P.J.W acknowledges EPSRC funding (EP/L016087/1).

## Author contributions

A.D.S. and M.Z. contributed equally. G.S.K.S., A.D.S. and M.Z. conceived the manuscript. A.D.S, M.Z, A.C. and G.F. prepared protein for experiments. A.D.S. and M.Z. performed kinetic aggregation assays. G.F. and A. DS. performed NMR. M.Z. and N.S. performed HDX-MS. R.M. performed native nano ESI-MS and native nano ESI-IM-MS. A.D.S. performed AFM experiments aided by I.M. P.J.W. performed preliminary FTIR experiments not included in the final paper. A.D.S, M.Z, G.F, R.M, J.J.P. and A. DS. analysed data. A.D.S, M.Z, R.M, J.J.P, H.L, A. DS., F.S and G.S.K.S. contributed to paper writing. All authors have given approval to the final version of the paper. All raw data is available on request and in the Cambridge University Repository.

## Competing interests

The authors declare no competing interests.
