## [Peer Review File · Nature Communications]

Reviewers' comments:

Reviewer #1 (Remarks to the Author):

In their manuscript, Stephens and Zacharopoulou et al. have conducted a study on alpha synuclein (aSyn), offering insights into the relationship between the N and C-terminal interactions that lead to different aggregation-prone and non-aggregation-prone behaviors. In particular, the authors have applied a series of structural techniques such as NMR, native and HDX-MS and different biophysical assays to a host of aSyn mutants to identify changes in conformational behavior upon phosphorylation, mutation and calcium binding. This is an extensive and interesting study and the authors observe interesting correlations in their data allowing them to propose a model that describes how the exposure of the N-terminal aSyn is sufficient to drive aggregation.

General concerns:

1. Overall the image of figures featured in the manuscript are not sufficient high. Cartoon schematics featured in Figure 1b and 7 could be improved by adding text labels and expanding on the detail of the schematic. For example, the text on lines 75-79 detail the role of electrostatic interactions between the N and C-terminus of aSyn and list NMR, MS and HDX-MS techniques but are underwhelmingly represented by Figure 1b. A full length NMR structure of aSyn is available as PDB 1XQ8. While this model likely does not encapsulate the extent of the aSyn conformational space in solution, it could be incorporated into Figure 1 for readers to better visualize what the aSyn structure may look like.

2. The authors appear to have ignored all currently available structural representations of aSyn, yet many different conformations are available from NMR, X-ray and cryo-EM techniques.

HDX-MS

The authors utilise HDX-MS experiments in order to identify regions of structural change in aSyn upon mutation or calcium binding. The results have been presented using "difference plots" at peptide resolution and the authors have described their results in terms of statistically significant changes in the N, NAC and C-terminal regions.

3. The authors observe deprotection in the N-terminus of A53T aSyn when calcium is added. They state that "for A53E aSyn (Figure 5f) no significant differences can be identified, neither at the NAC region nor at the N-terminus, upon calcium binding" (lines 274-275). However in Figure 5f, two statistically significant deprotected peptides can be seen in the N-terminal region. The same error is carried forward in the discussion section (line 450).

4. HDX-MS "difference plots" require some re-working to better display the data. Axis labels are too small, as are the residue numbers and state labels. The x-axis should be re-made to show the identity of each peptide. Alternatively the data may be presented in the also commonly used "Woods plot" format which uses a residue axis. In the current plot, there is no way to tell the length or start and end residues of each peptide. The panel labels should be made more obvious and moved to be above the panels rather than below. Figure S2 should be made larger and cut into two rows. Resizing the figure can be done in DynamX software.

5. Include in the HDX-MS methods section the filtering parameters used in PLGS and DynamX.

6. Include the significance values calculated for each differential peptide comparison.

7. Authors should visualize and present their uptake protection/deprotection data on the PDB of aSyn as mentioned in point 1 and also show the locations of mutations on the structure.

8. Could the authors explain why they have opted to not display peptides of aSyn which contain the mutations, while leaving the error trace visible?

ESI-IM-MS

The authors use ESI-IM-MS to characterize differences between conformations of aSyn in the gas phase. The data for the 8+ charge state is presented, of which the authors have divided into five distinct conformational groups of aSyn.

9. The text currently reads "WT aSyn was found to have four main conformational distributions at the 8+ charge state. The main four CCS values at 8+ was subdivided into regions labelled A-D, with region E containing ... ". It is somewhat confusing to introduce four conformations but then to list five regions.

10. Figure S12 displays the CCS distribution heat maps for charge states 5-17+ of different variants of aSyn. The authors may want to explain why they have focused on describing the "four" conformations of the 8+ state when additional low CCS conformations are observed at lower charge states. Could they justify why it is more relevant to study the 8+ charge state in particular?

11. In fact it appears that at least three additional low CCS conformations of gas phase aSyn be seen across 5-7+ charge states, with two co-populated conformations at 5+. The CCS corresponding to the lowest charge state (5+) of aSyn represents the least coulombically perturbed conformation. Upon addition of calcium to aSyn variants, the authors observe a shift in the distribution towards lower CCS in the 8+ charge state. Is this shift also visible in the 5+ charge state?

12. Include how many repeats were carried out for the IM-MS measurements and report errors associated with the area under curve measurements for the arrival time distributions.

13. Overall it is somewhat unclear what the ESI-IM-MS experiments add to the study. The authors state that "these findings thus highlight that monomeric aSyn can manifest in a variety of conformations" however is this not already obvious from the fact that the aSyn is an intrinsically disordered protein, not to mention that gas phase measurements of aSyn have already been reported previously (ref 44-46)? Being a gas phase measurement, the authors have also not explored the relevance of their IM-MS observations on the solution conformations of aSyn.

14. The current layout of Figure 6b can be improved by adding the aSyn variant names to the x-axis instead of 1-7 – which is confusing. Please also ensure that the tick labels are of consistent font and size throughout the figures to improve the legibility of the data.

15. Could the authors clarify what they mean by "upon calcium addition, the conformational ensemble shifted towards favoring compacted structures in all aSyn mutants as the C-terminus collapses due to charge neutralization upon binding to calcium", particularly the notion of C-terminus collapse?

Minor comments:

- Title: "alpha-synuclein" or "alpha synuclein"?
- Suggest changing the first sentence of the abstract to "As an intrinsically disordered protein, monomeric alpha synuclein (aSyn) occupies a large conformational space."
- Line 29: "re-lease"
- The molecular weight of aSyn is missing from the introduction
- The "aar" abbreviation is unnecessary, refer to as regions or residues, e.g. line 54: "the N-terminus, residues 1-60, which is .."
- Line 63: "wild-type (WT)"
- Line 79: add review ref 6 to the preceding list of references and remove "for a review"

- Line 81: change to "leading to differences in the aggregation propensity of aSyn"
- Line 90: change to "whether it enhances or retards aggregation"
- Line 178: expand "SI" to "Supplementary Information" and place in parentheses
- Line 184: add general explanation of "ThT-based kinetic assays" prior to explaining results.
- Line 221: "com-pared"
- Include a summary of the HDX data as suggested by Masson et al. Nature Methods, DOI: <https://doi.org/10.1038/s41592-019-0459-y>
- Line 324 is missing a reference to the supplementary discussion of Figure S11.
- Line 628: quoted redundancy of aSyn is inconsistent with figure.

Reviewer #2 (Remarks to the Author):

Please see attached file

Reviewer #3 (Remarks to the Author):

In this manuscript, Stephens et al. investigate by a variety of biophysical and structural methods how certain changes in the conformational ensemble of monomeric alpha-synuclein correlate with its tendency to form amyloid fibrils. They employ several variants of alpha-synuclein, most notably a phosphorylated and a reduced charge variant, but in addition also study the known disease-related variants. In addition, the authors use the addition of calcium in order to induce significant conformational changes in the disordered ensemble. Overall, this work is thoroughly performed, addresses an important question and uses mostly state of the art technology. Overall, I favour publication in Nature Communications, but I have a range of remarks that I would like the authors to comment on/address.

1) It is curious that Ca should bind cooperatively to alpha-synuclein, given that the negative charge, that drives the binding, decreases with every Calcium ion that binds. Therefore, one would expect negative cooperativity, rather than positive cooperativity. The authors should at least comment on this unexpected finding. Also, some of the fits to the cooperative model are not great. Furthermore, the authors state that calcium affinity of asyn does not correlate with its aggregation in the presence of Calcium. However, Calcium affinity would only be expected to correlate with aggregation rate under conditions where the binding sites are not fully saturated. Experiments at different degrees of calcium binding (i.e. at different Ca concentrations) would be required to be able to say something about that. Under the conditions where the authors perform the experiments, together with the reported binding affinities, one can assume that all Ca binding sites are saturated and hence no correlation with the affinity is expected. Lastly, I would like to hear whether the authors have any evidence for or against calcium binding in the sequence region where the majority of the familial mutants are situated? After all, the different familial mutants change the local charge there quite significantly (charge difference between E46K and WT is 2 units, and also A53E and G51D may change the local calcium affinity). If there is any evidence of, even weak, binding, then the argument of the authors will have to be extended.

2) I am rather skeptical of the FT-IR data. I don't see what conclusions should be drawn from the investigation of lyophilised protein, that are supposed to be valid for what happens in solution. In addition to that, everybody who has done protein lyophilisation before knows that the appearance of the dry protein can be quite variable, from dense to "fluffy", probably dependent on the exact lyophilisation conditions. Before being able to analyse differences in lyophilised protein between different variants, I would like to see that several repeats of the same protein are identical. If the authors cannot provide that, I would recommend removing the FT-IR data, it does not add anything to the story.

Actually, lyophilisation of alpha-synuclein has been shown to induce its oligomerisation, so the authors are adding an extra layer of complexity, which does not help, rather the opposite.

3) The same holds in my opinion for the ion mobility MS data. It is rather strange to try and interpret the conformational states of a +8 state of alpha-synuclein in the gas phase in the context of the rest of the manuscript, where it is shown that already single charge changes can lead to significant changes in the conformational ensemble in solution...

Again, this data does not add anything to the story. It seems like the authors are preparing another manuscript (they cite it somewhere as being "in preparation"), where they also look at the effects of singly charged cations. In my view the ion mobility data would be better valued if it were removed from this current manuscript and discussed with the additional data the authors have on the binding of other types of ions.

4) The ThT kinetics are rather variable. This should be highlighted a bit more and the conclusions based on the kinetics toned down. Also, in Figure 4 a) and b), individual repeats should be shown, rather than average curves. It is mathematically not very meaningful to average such highly variable curves, which not only show different lag times, but also different final fluorescence intensities.

5) The authors state that "It remains to be determined whether these different structures of monomeric aSyn can be isolated"

This seems very unlikely, given that the authors themselves say that aSyn forms an ensemble of rapidly interconverting structures. These structures probably equilibrate on a time scale many orders of magnitude faster than any purification experiment.

6) Finally, I think the authors should also discuss that what really matters for the aggregation is the energy landscape of the protein in contact with other protein molecules, rather than the energy landscape of the isolated monomer. The energy landscape of two or more molecules is, by definition, different from that of an isolated monomer. Therefore there is always the question how many conclusions about the energy landscape of intermolecular interactions can be drawn from studies of the energy landscape of isolated monomers. As soon as two monomers encounter each other, conformations may be accessible that are inaccessible for the isolated monomers.

Therefore, while it is possible to correlate conformations of the monomer with aggregation rates, such correlations do not imply any causality. This type of argument is not often made in these types of studies, but in my view is crucial.

Reviewer #2 (Remarks to the Author):

The paper by Stephens et al describes Ca²⁺ binding to WT alpha-synuclein aS and variants. The authors interpret the effects of high Ca²⁺ concentrations in terms of perturbations of long-range interactions within the aS conformational ensemble. I found these interpretations unconvincing because alternative mechanisms and controls were not considered and since some important experimental variables are not given.

Major points:

1. Much of the evidence for long-range interactions in the paper are NMR chemical shift perturbation however since the chemical shift is not a direct measurement of distance between NMR nuclei these data are at best indirect. It would have been more convincing to investigate long-range interactions or solvent accessibility of parts of aS by paramagnetic approaches: either paramagnetic relaxation enhancement (PRE) using paramagnetic-tagged aS or a solvent paramagnetic probe such as Mn²⁺ or TEMPOL to probe shielding from solvent.

- The Ca²⁺ concentration of 4.2 mM for the NMR experiments seems very large. This is 20x larger than the 200 μM aS concentration and 10 to 100 x larger than the K_d values calculated in Fig. S7 (depending on the model). At these large Ca²⁺ concentrations can the authors rule out weak Ca²⁺ binding or 'salting-out' effects could be causing the CSPs at the N-terminus? (a control could be to look for CSPs in an N-term fragment with or without Ca²⁺). Certainly Asp, Glu and phosphate would be suitable ligands for Ca²⁺ but at high enough concentrations other groups from the N-terminus including the backbone could bind. In this regard it should also be noted there are a number of Asp and Glu in the N-term aar 1-100 of aS. How specific are the effects to Ca²⁺ as opposed to other salts such as NaCl or spermine?
- The Authors claim the CSPs are due to the perturbation of "long-range" interactions in monomeric aS. However, could large Ca²⁺ concentrations cause aggregation of aS together with artifactual CSPs? This could be easily checked to see if the Ca²⁺-induced CSPs depend on protein concentration (e.g. do the measurements at 1 mM and 10 μM aS).
- For the primary HSQC data in Fig. 2C, only aar from 113 to 137 in the C-term are labeled so it's difficult to judge the magnitudes of the purported CSPs for the NAC and the N-term regions 1-10 and 70-80. The purported CSPs for 1-10 and 70-80 in Fig. 2A are also unconvincing as they are close to the noise level of the plots.
- How were the NMR spectra referenced? Since the authors claim that with D121A everything is shifted more in the presence of Ca²⁺. There is no information given how either ¹H or ¹⁵N shifts were referenced.
- How was temperature maintained and verified for the NMR experiments? HN amide peaks are sensitive to temperature (e.g. temperature factors) and in Fig. 2C for D121A the spectrum in the presence of Ca²⁺ seems to have amide (¹H_N)

resonances shifted systematically upfield (see Ser/Thr region), whereas the mechanism proposed by the authors makes it unlikely that shifts would be in a systematic direction.

- What was the pH of the samples, were they buffered, and where the sample pH values the same for WT and the mutants? Was it verified that the pH did not change after addition of Ca²⁺ (due to possible acid base impurities in the Ca²⁺ sample or sample handling)?

2. For the HX experiments I do not understand the composition of the quench buffer given in lines 619-620 as: (100 mM Tris, 8 M Urea, pH 2.45 at 0 °C). How is 100 mM Tris a buffer at pH 2.45? My concern is that HX is pH-dependent (base and acid catalyzed) and if the quench buffer is as stated it may result in different HX occupancies if the samples are not buffered.

- Differences between mutants and WT in MS Fig 3 where the mutations have no effect are not that much smaller than difference without and with Ca²⁺ (except for WT where the effect of Ca²⁺ does seem to be larger? Also if mutants have no effect why are all the HX changes consistent. Always negative in Fig. 3a, for example?

- Authors attribute changes in HX to changes in H-bonding or salt exposure but salt concentration (e.g. high Ca²⁺ concentration) could also affect HX. Also in principle since it is OH⁻ catalyzed HX could be affected by a change in charge (e.g. charge content for the C-term).

3. In Supplementary Fig. 7A the K_d values are in the 70-90 μM range. The aS concentration is given as 200 μM and the lowest Ca²⁺/aS ratios on the X-axis appear to be around 3 implying the lowest Ca²⁺ concentration was 600 μM. Why are the K_d values lower than any of the Ca²⁺ concentrations?

Minor points:

- I am not sure why the methods section mentions purification of ¹⁵N/¹³C-labeled samples since only ¹H-¹⁵N correlation spectra are shown or discussed in the main text. What was the ¹³C there for?

- It should be stated that the CS differences in Fig. 2 are absolute values because there are no negative differences. Also, why is the mutation site shown off-scale?

- for Supplementary Fig. 7B there are no units given for the K_d values. Presumably these are mM like they are in A.

- Maybe the authors should comment on why the HX was done by MS rather than NMR? Both methods could be used to measure HX and it would be useful to know if the methods agreed. If MS has advantages over NMR for this case the authors should

comment on this.

- line 517 - I believe the color for the C-term should be red not blue.

We would like to thank the reviewers very much for their insightful comments. They have been very helpful and we feel that our manuscript has been significantly improved as a result of the comments.

Reviewer #1 (Remarks to the Author):

General concerns:

1. Overall the image of figures featured in the manuscript are not sufficient high. Cartoon schematics featured in Figure 1b and 7 could be improved by adding text labels and expanding on the detail of the schematic. For example, the text on lines 75-79 detail the role of electrostatic interactions between the N and C-terminus of aSyn and list NMR, MS and HDX-MS techniques but are underwhelmingly represented by Figure 1b. A full length NMR structure of aSyn is available as PDB 1XQ8. While this model likely does not encapsulate the extent of the aSyn conformational space in solution, it could be incorporated into Figure 1 for readers to better visualize what the aSyn structure may look like.

We have improved Figure 1b and 7 and now show cartoon conformations that represent intramolecular interactions which have been previously identified in solution conditions by NMR and MS. However, as monomeric aSyn is an intrinsically disordered protein there are no PDB structures of the monomeric protein available. The 1XQ8 structure suggested by the reviewer represents aSyn in a micelle bound form which is known to be significantly different to the solution form of monomeric aSyn.

2. The authors appear to have ignored all currently available structural representations of aSyn, yet many different conformations are available from NMR, X-ray and cryo-EM techniques.

The current structures available online from NMR, X-ray and cryo-EM techniques do not represent monomeric aSyn. Indeed, we wanted to address whether changes in the monomeric protein influence the aggregation propensity of the protein, something which has not been addressed previously.

We have added 'It remains to be determined whether the structure of the initial monomer subsequently influences the tertiary structural fate of aSyn or influences fibril structure.' to the discussion, lines 457-459.

3. The authors observe deprotection in the N-terminus of A53T aSyn when calcium is added. They state that "for A53E aSyn (Figure 5f) no significant differences can be identified, neither at the NAC region nor at the N-terminus, upon calcium binding" (lines 274-275). However in Figure 5f, two statistically significant deprotected peptides can be seen in the N-terminal region. The same error is carried forward in the discussion section (line 450).

To be more precise we have now altered the text in the results to: 'In contrast, for A53E aSyn (Figure 6f, Table S12) there are mostly no significant differences at the NAC region or at the N-terminus (with the exception of peptides 55-89 and 55-94), upon calcium binding. Lines 317-319.'

Altered text in discussion to: 'No such deprotection was observed for the aSyn mutant A53E, where no significant differences were observed upon calcium binding, except for two small peptides as shown in Fig. 6f.'

4. HDX-MS “difference plots” require some re-working to better display the data. Axis labels are too small, as are the residue numbers and state labels. The x-axis should be re-made to show the identity of each peptide. Alternatively the data may be presented in the also commonly used “Woods plot” format which uses a residue axis. In the current plot, there is no way to tell the length or start and end residues of each peptide. The panel labels should be made more obvious and moved to be above the panels rather than below. Figure S2 should be made larger and cut into two rows. Resizing the figure can be done in DynamX software.

The figures have been amended, the peptides are now labelled, and the panel labels, axis labels, residue numbers and state labels are now larger and hopefully more legible. This figure is now Supplementary figure S11 and was resized and split into two parts.

5. Include in the HDX-MS methods section the filtering parameters used in PLGS and DynamX.

Thank you for this comment. The text in the method section has now been updated to include the different parameters: ‘The following parameters were used to filter the quality of the peptides: minimum and maximum peptide sequence length of 4 and 50, respectively, minimum intensity of 5000, and minimum products per amino acid of 0.2. All spectra generated from the peptides were examined and only peptides with high-quality spectra and a high signal to noise ratio were used for data analysis.’ Lines 614-618.

6. Include the significance values calculated for each differential peptide comparison.

The p-values for the peptide comparisons were calculated using an unpaired t-test, the alpha value was set to 1% ($p < 0.01$). All data have been added to the supplementary information, Tables S1-S5, S8-S12.

7. Authors should visualize and present their uptake protection/deprotection data on the PDB of aSyn as mentioned in point 1 and also show the locations of mutations on the structure.

It would be great to plot protection/deprotection data on a PDB structure, but unfortunately there are no structures of monomeric aSyn available, as discussed above.

8. Could the authors explain why they have opted to not display peptides of aSyn which contain the mutations, while leaving the error trace visible?

We agree with the reviewer and the error trace has been removed from the graphs.

ESI-IM-MS

9. The text currently reads “WT aSyn was found to have four main conformational distributions at the 8+ charge state. The main four CCS values at 8+ was subdivided into regions labelled A-D, with region E containing ... “. It is somewhat confusing to introduce four conformations but then to list five regions.

We agree with the reviewer and have now amended the text to read ‘The CCS values at 8+ were subdivided into five regions labelled A-E, Regions A-D contained the most populated conformations, with region E containing the least abundant and most compact conformations.’

And 'The familial mutations also display five co-existing conformational distributions, again with regions A-D being the most populated (Figure S16, S17). '

This section is now in the SI.

10. Figure S12 displays the CCS distribution heat maps for charge states 5-17+ of different variants of aSyn. The authors may want to explain why they have focused on describing the “four” conformations of the 8+ state when additional low CCS conformations are observed at lower charge states. Could they justify why it is more relevant to study the 8+ charge state in particular?

We agree with the reviewer and have now clarified the issue in the text. In the SI we have added an explanation for the choice of charge state: 'The 8+ charge state was chosen because it is one of the intermediate charge states which resembles conformations present in solution as determined by NMR (Allison, Rivers, Christodoulou, Vendruscolo, & Dobson, 2014); this state has multiple, clearly defined CCS distributions which reflect physiologically relevant states.' Page 28 of SI.

We have also added CCS data (new Figure S18) for the other charge states across the aSyn variants.

The next now reads, 'however there were no clear alterations in the distribution of conformations between the mutants and WT aSyn in the presence or absence of calcium or at other charge states (Figure S16 - S18, Table S6, S7).' Lines 293-295

11. In fact it appears that at least three additional low CCS conformations of gas phase aSyn be seen across 5-7+ charge states, with two co-populated conformations at 5+. The CCS corresponding to the lowest charge state (5+) of aSyn represents the least coulombically perturbed conformation. Upon addition of calcium to aSyn variants, the authors observe a shift in the distribution towards lower CCS in the 8+ charge state. Is this shift also visible in the 5+ charge state?

We agree with the reviewer, and we have now added CCS data for all charge states when aSyn is bound to 2 Ca²⁺ in Figure S19. We also observe compacting effects for the 5+ charge state (as well as the 6+ until around 9+ or 10+) but the intensity in the m/z spectrum of the 5+ is already quite low, so the intensity of peaks in the Ca²⁺ bound state are even lower. This is the reason we do not have data for the 5+ charge state for every mutant, as the intensity is too low to get a decent ion mobility drift time profile out of it.

12. Include how many repeats were carried out for the IM-MS measurements and report errors associated with the area under curve measurements for the arrival time distributions.

We apologise for this miss. This information and data have now been added to Table S6 and S7: 'p values were calculated from the average of three replicates of mutant aSyn and compared to the average of three replicates of WT aSyn'

13. Overall it is somewhat unclear what the ESI-IM-MS experiments add to the study. The authors state that “these findings thus highlight that monomeric aSyn can manifest in a variety of conformations” however is this not already obvious from the fact that the aSyn is an intrinsically disordered protein, not to mention that gas phase measurements of aSyn have already been reported previously (ref 44-46)? Being a gas phase measurement, the authors have also not explored the relevance of their IM-MS observations on the solution conformations of aSyn.

The IM-MS experiments confirm that there is more than one conformation present of monomeric aSyn in the gas phase. We do though agree that it is currently unclear how IM-MS measurements of IDPs in the gas phase can be related to proteins in solution. We have thus decided to move this figure and its discussion to the SI.

14. The current layout of Figure 6b can be improved by adding the aSyn variant names to the x-axis instead of 1-7 – which is confusing. Please also ensure that the tick labels are of consistent font and size throughout the figures to improve the legibility of the data.

Thank you for this comment. When we add the variant names to the axis it makes it very hard to read as the text is small due to the amount of columns, instead the use of numbers makes it easier to see the text clearly.

15. Could the authors clarify what they mean by “upon calcium addition, the conformational ensemble shifted towards favoring compacted structures in all aSyn mutants as the C-terminus collapses due to charge neutralization upon binding to calcium”, particularly the notion of C-terminus collapse?

We have now clarified this in the text, which is found in the SI page 28; Upon addition of calcium, compacted conformations are favoured in all aSyn variants, particularly conformation C, as indicated by more populated low CCS values (Figure S16a, +2Ca²⁺). There are currently two possible explanations for the observed compaction of aSyn upon calcium binding, either the C-terminus collapses due to charge neutralisation removing the electrostatic repulsion, which has also been observed for Mn²⁺ and Co²⁺ binding (Wongkongkathep, P, et al., J. Am. Soc. Mass Spectrom (2018), 29 (9)) or that the gas phase induces compaction upon specific divalent cation binding which leads to charge neutralisation and electrostatic interactions being favoured, which may not occur in solution (Han, J. Y et al., Sci. Rep. (2018), 8 (1).

Minor comments:

- **Title: “alpha-synuclein” or “alpha synuclein”?** – We have changed to hyphenated throughout the text
- **Suggest changing the first sentence of the abstract to “As an intrinsically disordered protein, monomeric alpha synuclein (aSyn) occupies a large conformational space.”** – We have now made this change.
- **Line 29: “re-lease”** – changed
- **The molecular weight of aSyn is missing from the introduction** – We have added in line 53 ‘Monomeric aSyn is 14.46 kDa and has three characteristic main regions;’
- **The “aar” abbreviation is unnecessary, refer to as regions or residues, e.g. line 54: “the N-terminus, residues 1-60, which is ..”** – We have removed ‘aar’ and replaced with residues.
- **Line 63: “wild-type (WT)”** – added
- **Line 79: add review ref 6 to the preceding list of references and remove “for a review”** – We have added the reference to the list and removed ‘for a review’ from text.
- **Line 81: change to “leading to differences in the aggregation propensity of aSyn”** – changed
- **Line 90: change to “whether it enhances or retards aggregation”** – changed
- **Line 178: expand “SI” to “Supplementary Information” and place in parentheses** – changed

- **Line 184: add general explanation of “ThT-based kinetic assays” prior to explaining results.** – added ‘Upon binding to β -sheet rich fibrillary structures the molecule ThT emits fluorescence which provides a read out for the kinetics of aSyn aggregation.’
- **Line 221: “com-pared”** – removed
- **Include a summary of the HDX data as suggested by Masson et al. Nature Methods, DOI: <https://doi.org/10.1038/s41592-019-0459-y>** -We have added the HDX data summary as per *Masson et al* in a supplementary Excel file
- **Line 324 is missing a reference to the supplementary discussion of Figure S11.** – We have moved this to SI.
- **Line 628: quoted redundancy of aSyn is inconsistent with figure.** – The redundancy was altered in Methods to match Figure S2 (now S11), redundancy of 5.38.

Reviewer #2 (Remarks to the Author):

Major points:

Much of the evidence for long-range interactions in the paper are NMR chemical shift perturbation however since the chemical shift is not a direct measurement of distance between NMR nuclei these data are at best indirect. It would have been more convincing to investigate long-range interactions or solvent accessibility of parts of aS by paramagnetic approaches: either paramagnetic relaxation enhancement (PRE) using paramagnetic-tagged aS or a solvent paramagnetic probe such as Mn²⁺ or TEMPOL to probe shielding from solvent.

We understand where the concern from the reviewer is coming from. However, we have designed our experiments in the way they are described for the following reasons: a) we wanted to perform NMR experiments in conditions that were as native as possible in order to probe the native conformation of aSyn in solution, as differences between the aSyn mutants were expected to be small. Addition of mutations such as the addition of a cysteine to the protein sequence in order to introduce a probe for PRE would significantly distort the monomer structure. Furthermore, the addition of a probe, which is usually hydrophobic, will again have interfered with the native intramolecular interactions of aSyn. Similarly, addition of Mn²⁺ will likely lead to direct binding of aSyn and alter the conformational properties of the protein. Similarly, TEMPOL is also a hydrophobic molecule that tends to stick to hydrophobic patches and in a protein like aSyn would likely to alter the conformational phase space, as IDPs are very sensitive to their surrounding environment and hydration levels. Since we were aware of the indirect nature of our measurements we decided to combine our NMR analysis with a complementary technique such as HDX-MS to investigate the solvent exposure of aSyn and to investigate long-range interactions, since NMR data most likely probe shorter range interactions.

The Ca²⁺ concentration of 4.2 mM for the NMR experiments seems very large. This is 20x larger than the 200 μM aS concentration and 10 to 100 x larger than the K_d values calculated in Fig. S7 (depending on the model). At these large Ca²⁺ concentrations can the authors rule out weak Ca²⁺ binding or 'salting-out' effects could be causing the CSPs at the N-terminus? (a control could be to look for CSPs in an N-term fragment with or without Ca²⁺). Certainly Asp, Glu and phosphate would be suitable ligands for Ca²⁺ but at high enough concentrations other groups from the N-terminus including the backbone could bind. In this regard it should also be noted there are a number of Asp and Glu in the N-term aar 1-100 of aS. How specific are the effects to Ca²⁺ as opposed to other salts such as NaCl or spermine?

In order to address the reviewer's concerns, we now have added the full data set for the calcium titration from 200 μM to 4.2 mM to the SI, Figure S4. As the Figure shows, we see no CSP and therefore no calcium binding at the N-terminus of aSyn. Represented in Figure 2 of the main manuscript are the CSPs for the final concentration of calcium used in the titration. Note, in our previous publication we have used 6 mM calcium and found saturation to occur ~3.6 mM where no more CSPs occurred.

We have further performed additional experiments in the presence of NaCl showing that there is no peak motion in the presence of 4 mM NaCl, ruling out the salting out effect (new Figure S5). We then increased the concentration of NaCl to 150 mM and observed peak shifts across the sequence when compared to the absence of salt, showing no specific interaction of aSyn with monovalent ions (Figure R1). We then performed an experiment with 150 mM NaCl and the addition of 4.2 mM and 8.4 mM CaCl₂ and observed peak shifts at the C-terminus even in the presence of salt (shown below in Figure R2 for 150 mM NaCl and 8.4 mM CaCl₂), indicating that the aSyn-calcium interaction is specific at the C-terminus.

[Redacted]

Figure R1. [Redacted]

[Redacted]

Figure R2. [Redacted]

The Authors claim the CSPs are due to the perturbation of “long-range” interactions in monomeric aS. However, could large Ca²⁺ concentrations cause aggregation of aS together with artifactual CSPs? This could be easily checked to see if the Ca²⁺-induced CSPs depend on protein concentration (e.g. do the measurements at 1 mM and 10 μM aS).

We do not think there is evidence of any aggregation of aSyn during these measurements at 10°C as this would also lead to significant by peak broadening, which is not evident in our data. However, to further confirm this, we have performed additional experiments with 200 μM and 10 μM aSyn in the presence of 4.2 mM Ca²⁺, resulting in overlapping 1H-15N HSQC and no aggregation at 200 μM (see Figure R3 below).

[Redacted]

Figure R3. [Redacted]

For the primary HSQC data in Fig. 2C, only aar from 113 to 137 in the C-term are labeled so it's difficult to judge the magnitudes of the purported CSPs for the NAC and the N-term regions 1-10 and 70-80. The purported CSPs for 1-10 and 70-80 in Fig. 2A are also unconvincing as they are close to the noise level of the plots.

We agree it is difficult to determine whether the CSPs from 1-10 and 70-80 are above noise level therefore we described them as 'possible' CSPs. We have, however, now removed this sentence to reduce ambiguity. We have added additional labelling to the HSQC figure, now found in the supplementary figure S6 and S7.

How were the NMR spectra referenced? Since the authors claim that with D121A everything is shifted more in the presence of Ca²⁺. There is no information given how either ¹H or ¹⁵N shifts were referenced.

We have used external referencing with the DSS molecule (direct referencing on ¹H and indirect referencing for ¹⁵N). We then performed explicit referencing when repeating aSyn D121A measurements and found the spectra to overlap perfectly.

How was temperature maintained and verified for the NMR experiments? HN amide peaks are sensitive to temperature (e.g. temperature factors) and in Fig. 2C for D121A the spectrum in the presence of Ca²⁺ seems to have amide (HN) resonances shifted systematically upfield (see Ser/Thr region), whereas the mechanism proposed by the authors makes it unlikely that shifts would be in a systematic direction.

The internal thermostat of our instrument is extremely robust and we have always made sure that the temperature equilibrium is reached before experiments are performed. The standard deviation of the temperature in our instrument, as measured using the methanol standard over a period of 4h, is 0.008 K (see figure R4 below).

Figure R4. Temperature measurement over 4 hours using methanol standard maintained at 283 K with 0.008 K sd.

As for the suggested upfield shift, we now show that this is not the case, by reporting the direct difference of ¹H and ¹⁵N chemical shifts as measured with and without Ca²⁺ (please see Figure R5 below). This plot shows a both positive and negative chemical shift differences, with specific trends in different regions of the protein.

Figure R5. Direct difference of ^1H (yellow) and ^{15}N (red) chemical shifts as measured with and without Ca^{2+} for D121A aSyn

What was the pH of the samples, were they buffered, and where the sample pH values the same for WT and the mutants? Was it verified that the pH did not change after addition of Ca^{2+} (due to possible acid base impurities in the Ca^{2+} sample or sample handling)?

For both NMR and HDX-MS measurements, which are very sensitive to pH changes, the pH has been tested and found to be the same before and after calcium addition. We have now added: ‘The pH of samples was measured before experiments were performed and also after calcium addition.’ to the methods section, lines 563-564.. Buffer conditions were also consistent throughout our measurements and are described in the text.

For the HX experiments I do not understand the composition of the quench buffer given in lines 619-620 as: (100 mM Tris, 8 M Urea, pH 2.45 at 0 °C). How is 100 mM Tris a buffer at pH 2.45? My concern is that HX is pH-dependent (base and acid catalyzed) and if the quench buffer is as stated it may result in different HX occupancies if the samples are not buffered.

It is standard practice in the HDX-MS community to use the same buffer chemical (in this case, Tris) for labelling and quench buffers, in large part to maintain protein solubility in a buffer known to be compatible with the protein(s) being studied. The pH of the quench buffer is precisely adjusted to pH 2.45 at 0° C and, furthermore, we have measured the pH of the quenched protein sample to be 2.47. If the pH of the quenched sample was outside the range (pH 2.45-2.55) where amide HDX is minimal between the acid and base-catalyzed mechanisms, as stated by the Reviewer, we would see significant back-exchange in our data, which we did not.

Differences between mutants and WT in MS Fig 3 where the mutations have no effect are not that much smaller than difference without and with Ca²⁺ (except for WT where the effect of Ca²⁺ does seem to be larger? Also if mutants have no effect why are all the HX changes consistent. Always negative in Fig. 3a, for example?

The difference in deuterium uptake between the calcium-bound and -unbound state was evaluated for each peptide by an unpaired Student t-test. We have now re-analyzed the data and found no significant differences ($p < 0.01$) when comparing between the different aSyn variants in the unbound state (e.g. pS129 vs WT, A53T vs WT), with very few peptide exceptions (please see updated figures 3 and 6).

We agree with the reviewer that it is highly likely that a point mutation or post-translational modification changes the conformation of aSyn (or the conformer distribution) compared to WT, which is, in fact, part of our hypothesis. As HDX-MS is a technique that averages across the conformational population, we believe that the conformational changes which are induced upon point mutation or phosphorylation are too small to be detected with our current technique, so we see no significant differences. Measurements with increased sensitivity are part of ongoing work in an attempt to resolve these small, but potentially functionally significant, changes. Nonetheless, here we were able to measure significant changes to the aSyn conformer population in response to calcium addition.

Authors attribute changes in HX to changes in H-bonding or salt exposure but salt concentration (e.g. high Ca²⁺ concentration) could also affect HX. Also in principle since it is OH⁻ catalyzed HX could be affected by a change in charge (e.g. charge content for the C-term).

Whilst the Reviewer is correct that the precise chemical composition – and not simply pH - affects hydrogen exchange rates, we are convinced that this does not contribute to any inconsistency with interpretation for the following reasons: (i) Since, upon calcium addition, we observe protection at the C-terminus where the expected binding site is and simultaneously deprotection at the N-terminus, we believe that the change in HDX is indicating a conformational change. (ii) Were HDX rates to alter in response to salt concentration, we would expect them to show differences in a systematic manner across all exchangeable sites and between aSyn states.

In Supplementary Fig. 7A the K_d values are in the 70-90 μM range. The aS concentration is given as 200 μM and the lowest Ca²⁺/aS ratios on the X-axis appear to be around 3 implying the lowest Ca²⁺ concentration was 600 μM. Why are the K_d values lower than any of the Ca²⁺ concentrations?

The lowest Ca²⁺/aSyn ratio of the Supplementary figure S8A is 1. This is more clearly visible in the panel B where the same data are plotted in logarithmic scale.

We do however agree that the K_d values in the panel A are too low, probably because of the imperfect fit. For this reason, we had decided to use a new model (the Hills equation, Panel B) leading to better fitting and different ranges of K_d (i.e. 460-670 μM).

Minor points:

- **I am not sure why the methods section mentions purification of 15N/13C-labeled samples since only 1H-15N correlation spectra are shown or discussed in the main text. What was the 13C there for?** – This batch of protein was also used for other experiments requiring 13C.
- **It should be stated that the CS differences in Fig. 2 are absolute values because there are no negative differences. Also, why is the mutation site shown off-scale?** – The mutation site when compared to WT gives very high ΔCS, as expected. Thus, in order to observe ΔCS in other regions of

the protein we show a zoomed in section. The full spectra are now shown in the SI for WT-D121A and WT-pS129.

- **for Supplementary Fig. 7B there are no units given for the Kd values. Presumably these are mM like they are in A.** – this is now amended
- **Maybe the authors should comment on why the HX was done by MS rather than NMR? Both methods could be used to measure HX and it would be useful to know if the methods agreed. If MS has advantages over NMR for this case the authors should comment on this.** –HDX-MS can sensitively measure the structural perturbations of a protein, and has several key advantages: the measurement is fast, more high-throughput and can assess rapid kinetic processes; sample consumption is low; there are few constraints on buffer conditions; and data is sub-molecular and can be up to single amino acid resolution.
- **line 517 - I believe the color for the C-term should be red not blue.** – The C-terminus is red. In the text we are referring to the N-terminus, which is blue.

Reviewer #3 (Remarks to the Author):

1) It is curious that Ca should bind cooperatively to alpha-synuclein, given that the negative charge, that drives the binding, decreases with every Calcium ion that binds. Therefore, one would expect negative cooperativity, rather than positive cooperativity. The authors should at least comment on this unexpected finding. Also, some of the fits to the cooperative model are not great.

We thank the reviewer for highlighting this, we will explain this in more detail in the text. We do not believe that the charge and electrostatic interactions are the main drivers for calcium binding, as we do not see a big change in Kd between the charge mutants, pS129 with an additional negative charge and D121A with a reduced negative charge. Furthermore, there is no difference in the number of calcium ions bound between D121A and WT aSyn, when it may be expected that D121A aSyn binds less calcium when it is less negatively charged. We instead believe that a conformational change, perhaps formation of a pocket, instead which may aid calcium binding. Therefore, positive cooperativity is possible when a conformation is formed that favours calcium binding. The fits for the models have been amended and the R values are close to 1, indicating a goodness of fit.

We have added to the text 'Furthermore, our results show that there is positive cooperativity for calcium binding to aSyn, which indicates that charge and electrostatic interactions are not the driving force for calcium binding, as this would lead to negative cooperativity. We argue that calcium binding leads to a conformational change, which consequently leads to a positive cooperativity.' Lines 372 – 376.

Furthermore, the authors state that calcium affinity of asyn does not correlate with its aggregation in the presence of Calcium. However, Calcium affinity would only be expected to correlate with aggregation rate under conditions where the binding sites are not fully saturated. Experiments at different degrees of calcium binding (i.e. at different Ca concentrations) would be required to be able to say something about that. Under the conditions where the authors perform the experiments, together with the reported binding affinities, one can assume that all Ca binding sites are saturated and hence no correlation with the affinity is expected.

As discussed above, we believe that it is the conformational change, not the number of calcium ions or binding affinity that drive aggregation propensity. Indeed, our data show that there is correlation between the N-terminus exposure and disruption of intra-molecular interactions and the aggregation propensity of pS129, D121A, A53E, A53T and WT aSyn.

Lastly, I would like to hear whether the authors have any evidence for or against calcium binding in the sequence region where the majority of the familial mutants are situated? After all, the different familial mutants change the local charge there quite significantly (charge difference between E46K and WT is 2 units, and also A53E and G51D may change the local calcium affinity). If there is any evidence of, even weak, binding, then the argument of the authors will have to be extended.

As addressed above for point 2 of reviewer 2, we do not observe CSPs at the N-terminus by NMR for WT, D121A or pS129 aSyn upon calcium binding. HDX-MS data presented in this paper show no protection at the N-terminus upon calcium binding to suggest that Ca²⁺ binds at N-terminus in WT, A53T and A53E aSyn. Furthermore, no other published papers studying calcium binding to aSyn have yet reported that calcium binds to the N-terminus of aSyn (Nielsen, et al., JBC (2001), Binolfi, et al.,

JACS (2006), Lu, et al., ACS Chem Neuro (2011), Han, et al., Sci Rep (2018)). To date the only ion found to bind at the N-term is Cu^{2+} , but this appears to be an anomaly for divalent cations.

2) I am rather skeptical of the FT-IR data. I don't see what conclusions should be drawn from the investigation of lyophilised protein, that are supposed to be valid for what happens in solution. In addition to that, everybody who has done protein lyophilisation before knows that the appearance of the dry protein can be quite variable, from dense to "fluffy", probably dependent on the exact lyophilisation conditions. Before being able to analyse differences in lyophilised protein between different variants, I would like to see that several repeats of the same protein are identical. If the authors cannot provide that, I would recommend removing the FT-IR data, it does not add anything to the story. Actually, lyophilisation of alpha-synuclein has been shown to induce its oligomerisation, so the authors are adding an extra layer of complexity, which does not help, rather the opposite.

We agree with the reviewer and have removed the data.

3) The same holds in my opinion for the ion mobility MS data. It is rather strange to try and interpret the conformational states of a +8 state of alpha-synuclein in the gas phase in the context of the rest of the manuscript, where it is shown that already single charge changes can lead to significant changes in the conformational ensemble in solution...

Again, this data does not add anything to the story. It seems like the authors are preparing another manuscript (they cite it somewhere as being "in preparation"), where they also look at the effects of singly charged cations. In my view the ion mobility data would be better valued if it were removed from this current manuscript and discussed with the additional data the authors have on the binding of other types of ions.

IM-MS data indicate that monomeric aSyn can be present in a variety of conformations. They further indicate that calcium significantly skews the conformational ensemble of the different aSyn variants. However, the technique is not yet capable to resolve clear structural differences between the aSyn variants and WT aSyn and it is currently not clear how these distributions of conformations in the gas phase relate to solution conditions.' We have thus moved the IM-MS to the supplementary information. We have also added a discussion on the relevance of the gas phase and the problems with resolution in the SI pages 20-28.

4) The ThT kinetics are rather variable. This should be highlighted a bit more and the conclusions based on the kinetics toned down. Also, in Figure 4 a) and b), individual repeats should be shown, rather than average curves. It is mathematically not very meaningful to average such highly variable curves, which not only show different lag times, but also different final fluorescence intensities. –

We agree that ThT assay kinetics can be variable, yet they are still widely used by the community. We see large differences in the familial mutant aggregation rates which we believe are true as we also measured the remaining monomer concentration and this data also reflects the aggregation state of the aSyn mutants as seen by ThT assays. We have now added plate repeat data to the SI, Figure S12, and all individual trace data are available in the Cambridge University repository.

5) The authors state that "It remains to be determined whether these different structures of monomeric aSyn can be isolated" This seems very unlikely, given that the authors themselves say

that aSyn forms an ensemble of rapidly interconverting structures. These structures probably equilibrate on a time scale many orders of magnitude faster than any purification experiment.

We have amended the text to make it clear that the monomer conformations would need to be fixed before isolation. 'It remains to be determined whether these different structures of monomeric aSyn can be isolated, potentially by cross-linking to stabilise structures, or whether they have different toxicity in cells.' Lines 477-479.

6) Finally, I think the authors should also discuss that what really matters for the aggregation is the energy landscape of the protein in contact with other protein molecules, rather than the energy landscape of the isolated monomer. The energy landscape of two or more molecules is, by definition, different from that of an isolated monomer. Therefore there is always the question how many conclusions about the energy landscape of intermolecular interactions can be drawn from studies of the energy landscape of isolated monomers. As soon as two monomers encounter each other, conformations may be accessible that are inaccessible for the isolated monomers. Therefore, while it is possible to correlate conformations of the monomer with aggregation rates, such correlations do not imply any causality. This type of argument is not often made in these types of studies, but in my view is crucial.

We think this is an interesting point which should be further highlighted and discussed in the text. We agree with the reviewer that the energy landscape of the protein in contact with other protein molecules is important but it is known from the literature that dimer conformations are more energetically unfavourable compared to the monomer (Urbanc, et al., Biophys J. 2004). Thus, the transition from an isolated monomer that is more or less stabilised by intramolecular interactions to a structure that has its N-terminus exposed, thereby permitting dimer formation, is of high interest. We believe that if a monomeric structure can be further stabilised in a way that it prevents dimer formation, we might be able to prevent fibril formation all along.

We have added to the discussion, 'Furthermore, it would be interesting to investigate whether these conformational changes and exposure of the N-terminus lead to a reduction of the energy barrier needed to be overcome for dimer formation, which may be crucial in determining whether aSyn becomes pathological or not.' Lines 479-482.

REVIEWERS' COMMENTS:

Reviewer #1 (Remarks to the Author):

The authors have made a good effort to address some of the comments raised and indeed the manuscript has been improved.

Reviewer #2 (Remarks to the Author):

I've looked over the revised MS by Stephens et al and the reviewer rebuttal. A lot of the missing experimental information in the original MS has now been included and some of my concerns have been addressed. I find it a bit odd that in the rebuttal figure R1 on the effects of 150 mM NaCl, many of the CSPs are of the same residues (121,126,137, 113, 119, 123, 131) and in the same direction as with 4 mM CaCl₂ (Fig. R3, Fig S6). Although some peaks like 129 change with Ca²⁺ but no NaCl. Another puzzling feature in Fig. R2 (while this experiment is elegant) is that Ca²⁺-induced CSPs seem to need 8.4 mM Ca²⁺ in the presence of physiological salt (150 mM NaCl). This seems like an extremely high Ca²⁺ concentration making me wonder about the biological relevance of the results. Were all the other experiments done in the absence of any added salt? The salt conditions should be stated in the paper.

Minor points:

- I don't understand how one of the biggest shifters in Fig S6 can be 121 (1H 7.8 ppm, 15N 118.9 ppm) if this is the D121A mutant according to the figure legend (and the same peak appears in WT in Fig. R2). Also, this peak appears in a wrong position for an Ala in a 1H-15N HSQC and is more consistent with an Asp. The same seems to be going on in Fig. R3 but that's not in the paper. I think something must be labeled improperly.
- I still don't think Tris should be called a "buffer" at pH 2.45 in line 607.

Reviewer #3 (Remarks to the Author):

The authors have addressed my comments satisfactorily and I recommend publication as is.

Reviewer #2 (Remarks to the Author):

I've looked over the revised MS by Stephens et al and the reviewer rebuttal. A lot of the missing experimental information in the original MS has now been included and some of my concerns have been addressed. I find it a bit odd that in the rebuttal figure R1 on the effects of 150 mM NaCl, many of the CSPs are of the same residues (121,126,137, 113, 119, 123, 131) and in the same direction as with 4 mM CaCl₂ (Fig. R3, Fig S6). Although some peaks like 129 change with Ca²⁺ but no NaCl.

We would like to thank the reviewer for further critical comments.

Related to the above comment by reviewer 2, we expect that NaCl and CaCl₂ have similar electrostatic effects when interacting as ions at the charged N and C-termini, and thus they may induce some similar patterns of CSPs. For Ca²⁺ there are, however, specific interactions at the C-terminus which are not observed in the presence of Na⁺. As demonstrated in Figure R2, Ca²⁺ specifically interacts with the C-terminus, even in the presence of high amounts of NaCl. For example, as pointed out by the reviewers, residue 129 may be part of the specific Ca²⁺ binding site, but not for Na⁺.

Another puzzling feature in Fig. R2 (while this experiment is elegant) is that Ca²⁺-induced CSPs seem to need 8.4 mM Ca²⁺ in the presence of physiological salt (150 mM NaCl). This seems like an extremely high Ca²⁺ concentration making me wonder about the biological relevance of the results. Were all the other experiments done in the absence of any added salt? The salt conditions should be stated in the paper.

The addition of salt does indeed decrease Ca²⁺ binding affinity due to competition for binding with the Na⁺ ions, but Ca²⁺ strongly outcompetes Na⁺, as shown by peak motions occurring even at 4.2 mM Ca²⁺ despite the presence of 150 mM of Na⁺. The experiment which was performed upon suggestion of the reviewer, shows that the calcium binding is specific to this ion. We performed the experiment using both 4.2 mM CaCl₂ and 8.4 mM CaCl₂, and we now also display the data for 4.2 mM which show, as in the case of 8.4 mM, the specific C-terminus binding of Ca²⁺ in the presence of NaCl.

The concentration of CaCl₂ used in this experiment is selected to probe the specificity of binding with aSyn at the protein concentrations required for NMR experiments (200 μM), the resulting aSyn:calcium ratio is 1:21 for 4.2 mM CaCl₂ and 1:42 for 8.4 mM CaCl₂. The resulting binding patterns are valid also at lower concentrations of both the protein and Ca²⁺. We are further investigating the influence of salt and calcium, but this is not in the remit of the current manuscript.

[Redacted]

Figure R1. [Redacted]

Minor points:

- **I don't understand how one of the biggest shifters in Fig S6 can be 121 (1H 7.8 ppm, 15N 118.9 ppm) if this is the D121A mutant according to the figure legend (and the same peak appears in WT in Fig. R2). Also, this peak appears in a wrong position for an Ala in a 1H-15N HSQC and is more consistent with an Asp. The same seems to be going on in Fig. R3 but that's not in the paper. I think something must be labeled improperly.**

We apologise, D121A aSyn was used in figures R1, R2 and R3, but not stated in the figure legend of R1 and R2.

- **I still don't think Tris should be called a "buffer" at pH 2.45 in line 607.**

We agree Tris is not 'buffering' at pH 2.45, we will call it Tris only in line 607.